

# Climate model Selection by Independence, Performance, and Spread (ClimSIPS) for regional applications

Anna L. Merrifield[1], Lukas Brunner[2], Ruth Lorenz[1], Vincent Humphrey[1], and Reto Knutti[1]

[1]Institute for Atmospheric and Climate Science, ETH Zurich, Zurich, Switzerland
[2]Department of Meteorology and Geophysics, University of Vienna, Vienna, Austria

**Correspondence:** Anna L. Merrifield (anna.merrifield@env.ethz.ch)

**Abstract.** As the number of models in Coupled Model Intercomparison Project (CMIP) archives increase from generation to generation, there is a pressing need for guidance on how to interpret and best use the abundance of newly available climate information. CMIP6 users seeking to draw conclusions about model agreement must contend with an "ensemble of opportunity" containing similar models that appear under different names. Those who used CMIP5 as a basis for downstream applications

must filter through hundreds of new CMIP6 simulations to find several best suited to their region, season, and climate horizon of interest. Here we present methods to address both issues, model dependence and model subselection, to help users previously anchored in CMIP5 to navigate CMIP6. In Part I, we refine a definition of model dependence based on climate output, initially employed in Climate model Weighting by Independence and Performance (ClimWIP), to designate discrete model families within CMIP5/6. We show that the increased presence of model families in CMIP6 bolsters the upper mode of the ensemble's

bimodal effective Equilibrium Climate Sensitivity (ECS) distribution. Accounting for the mismatch in representation between model families and individual model runs shifts the CMIP6 ECS median and 75th percentile down by $0.43°C$, achieving better alignment with CMIP5's ECS distribution.

In Part II, we present a new, cost-function minimization-based approach to model subselection, Climate model Selection by Independence, Performance, and Spread (ClimSIPS), that selects sets of CMIP models based on the relative importance

a user ascribes to model independence (as defined in Part I), model performance, and ensemble spread in projected climate outcome. We demonstrate ClimSIPS by selecting sets of three to five models from CMIP5/6 for European applications, evaluating the performance from the agreement with the observed mean climate, and the spread in outcome from the projected midcentury change in surface air temperature and precipitation. To accommodate different use cases, we explore two ways to represent models with multiple members in ClimSIPS, first, by ensemble mean and second, by an individual ensemble mem-

ber that maximizes midcentury change diversity within CMIP overall. Because different combinations of models are selected by the cost function for different balances of independence, performance, and spread priority, we present all selected subsets in ternary contour "subselection triangles" and guide users with recommendations based on further qualitative independence, performance, and spread standards. In CMIP6, we find that recommended subsets are populated primarily by members of several model families defined in Part I due to an inverse relationship between performance and independence. In CMIP5,

recommended subsets feature model combinations used in the European branch of the Coordinated Regional Downscaling Ex-



periment (EURO-CORDEX), suggesting the independence, performance, and spread metrics used in ClimSIPS are appropriate
for European applications in CMIP6 and beyond.

# 1 Introduction

Since its inception in 1995, the Coupled Model Intercomparison Project (CMIP) has guided the climate science community in a
coordinated effort to understand how climate variability and change are represented by coupled ocean-atmosphere–cryosphere–land
general circulation models (GCMs; Meehl et al., 1997, 2000; Taylor et al., 2012; Eyring et al., 2016). The backbone of inter-
national climate assessments (IPCC, 2021), CMIP's common experiments have generated a range of possible future climate
outcomes representative of a range of modeling strategies, socioeconomic decision-making, and inherent systemic internal
climate variability. Generation to generation, CMIP model archives have grown, due to the participation of new modeling cen-
ters and to the recognition that multiple realizations of a single model provide valuable estimates of uncertainty arising from
internal variability (e.g. Haughton et al., 2014; Deser et al., 2020; Maher et al., 2021). Though these larger multi-model ensem-
bles represent advancements in global coordination and uncertainty representation, they present interpretation and utilization
challenges for downstream users (Dalelane et al., 2018).

## 1.1 The composition of CMIP

Interpreting results derived from multiple CMIP models is complicated by the fact that CMIP is an "ensemble of opportunity";
the project assembles all available climate projections that adhere to its simulation guidelines (Knutti et al., 2010). This inclu-
sive strategy collects "best guesses" from modeling groups with the capacity to participate, with that capacity ranging from
long-running, well-funded climate model development programs to newly-available computational resources for running a ver-
sion of an existing climate model. While being inclusive, such ensembles of opportunity are not designed to be a representative
sample of multi-model uncertainty in the way most would envision. For example, one might consider a representative sample
of multi-model uncertainty to be a distribution put forth by a set of distinct climate models with different but plausible strate-
gies for simulating the Earth system, equally represented by a single model run. Further, each of those distinct models could
be represented by several runs that start from slightly different states (initial condition ensemble members) to reflect internal
variability and by several runs that differ by parameter values (perturbed physics ensemble members) to reflect parametric
uncertainty (Parker, 2013), with the same number of runs for each model to maintain equal representation.

In reality, though, CMIP6 features over 60 uniquely named models (and counting) while its predecessor CMIP5 featured
on the order of 40. Uniquely named models range in terms of representation within the ensemble, from a single model run to
several member perturbed physics ensembles to 50-member single model initial condition large ensembles. modeling centers
often contribute several versions of their base model under different names as well (Leduc et al., 2016); these variants differ by,
for example, the spatial resolution of some model components or biogeochemical cycling, which may influence their simulated
climate in ways that are difficult to anticipate. Adding further complexity, even uniquely named models from different modeling
centers fall along a spectrum of uniqueness. Different models share historical predecessors (Masson and Knutti, 2011; Knutti



et al., 2013), conceptual frameworks, and, in some cases, source code (Boé, 2018). An active field of research has developed to identify and manage these "hidden dependencies" through weighting or subselection of the broader CMIP archives (e.g. Bishop

and Abramowitz, 2013; Sanderson et al., 2015; Knutti et al., 2017), but open questions remain, particularly with regards to how best to determine dependence within multi-model ensembles (Abramowitz et al., 2019; Annan and Hargreaves, 2017).

Dependence is important to identify within multi-model ensembles because it undermines the notion that when different approaches converge to the same outcome, consensus suggests certainty or robustness (Parker, 2013). Robustness requires that (1) a hypothesized outcome is likely to be true, (2) available evidence supports the hypothesized outcome, and (3) the

hypothesized outcome holds even if some assumptions within the body of supporting evidence fail (Parker, 2011). As an example, that global temperatures rise with increasing greenhouse gas concentrations in the atmosphere is an hypothesized outcome in climate science that is likely to be true. Many independent lines of evidence support this hypothesized outcome (e.g. Foote, 1856; Mitchell, 1989; Sherwood et al., 2020) and climate models that make different assumptions about how the Earth system should be modelled do as well, increasing confidence. Finally, even if an assumption made in one climate model

is found to be incorrect, the hypothesized outcome is secure (or still likely to be true) due to the diversity of the supporting evidence (Staley, 2004).

One can then see how important different modeling approaches are to robustness statements that are made in a solely multi-model context. Different modeling approaches are the diversity in the supporting evidence required for both the confidence in and security of an outcome. Because CMIP is not systematically designed to equally sample different modeling approaches,

ensemble agreement could be coming from a diverse set of models or could simply be coming from the same (or similar) models supporting an outcome repeatedly (Pirtle et al., 2010). Redundant agreement reflects certainty in a particular model's outcome, but does not mean that that model's outcome is necessarily correct nor that we should be more confident in that outcome overall. Too many highly dependent entities within an ensemble clearly shift and/or narrow uncertainty estimates (Merrifield et al., 2020), so it is, therefore, crucial to systematically identify dependencies and evaluate how they affect distributional

statistics before statements about robustness or uncertainty are made.

One method that has been developed to ward against over-confident multi-model climate uncertainty estimates is Climate model Weighting by Independence and Performance (ClimWIP; e.g. Knutti et al., 2017; Lorenz et al., 2018; Brunner et al., 2019, 2020b). ClimWIP uses model output variables to identify (1) potential issues that preclude a model from successfully simulating a realistic future climate response (performance) and (2) similarities that suggest a model is a duplicate or close

relation of another in the ensemble (independence). Initial versions of ClimWIP-based performance and independence definitions on the same set of predictors, which lead to concerns about convergence to reality. The basic concern was that as models improved, their (valid) agreement towards an outcome would be interpreted as dependence and result in them being downweighted. To address this concern, separate sets of predictors were introduced to define performance and independence within ClimWIP to allow for a straightforward and universal definition of dependence in line with prior knowledge of model

origin (Merrifield et al., 2020).

In addition to providing an operational definition of dependence that can be used to contextualize CMIP-derived results, ClimWIP has the advantage of being available for general open use (Sperna Weiland et al., 2021) as part of the Earth System



Model Evaluation Tool (ESMValTool; Righi et al., 2020). In the first part of this study, we revisit and refine ClimWIP's definition of dependence using long-term, large-scale climatological "fingerprints" that enhance the spread between models and reduce internal variability. We show that distances between different climate models, versions of the same model, or even between initial condition members derived from climatological fingerprints delineate levels of dependence within CMIP more precisely than distances based on previous predictor sets. This allows us to better illustrate how ensemble composition has changed from CMIP5 to CMIP6 in low-dimensional projected space. These intermember distances also reveal the presence of broader "model families" within CMIP. In light of this, we determine a potential point of separation between model families and individual runs in CMIP5 and CMIP6 (henceforth CMIP5/6) and validate the resulting family designations using model metadata. Finally, to better understand how dependence may affect CMIP uncertainty estimates, we investigate how restricting representation to one "vote" per family constrains distributions of effective equilibrium climate sensitivity (ECS) (Gregory et al., 2004).

Sections 2 through 4 comprise Part I of this study. Section 2 details the CMIP5/6 base ensembles used throughout. In Section 3, refinements made to ClimWIP's dependence strategy for the purpose of defining model families are described. Model families designations are put forward in Section 4 and subsequently employed to introduce a one vote per family constraint on ECS in CMIP5/6.

## 1.2 CMIP for downstream applications

Understanding dependencies within a multi-model CMIP ensemble is only the first step to designing an ensemble suitable for a downstream climate service application (Dalelane et al., 2018). For many applications, using the entirety of a modern CMIP archive may be too computationally expensive. It has been widely assumed within the impact and regional modeling communities that a subset of several CMIP simulations will suffice for most tasks, provided the subset retains key characteristics of the larger selected-from ensemble such as spread (e.g. Evans et al., 2013; McSweeney and Jones, 2016; Christensen and Kjellström, 2020; Kiesel et al., 2020; Qian et al., 2021). For example, CMIP5-era guidance from the European branch of the Coordinated Regional Downscaling Experiment (EURO-CORDEX) recommended participants drive regional climate models with the largest available subset of general circulation models selected based on factors such as historical model quality, ECS, or the retention of full ensemble spread in projected climate change signals (Benestad et al., 2021). Conscientious of computational burden, CORDEX proposed a minimum subset of three CMIP5 models, NorESM1, MPI-ESM, and HadGEM2-ES, with a further two models, GFDL-ESM and EC-EARTH, recommended as secondary alternatives (CORDEX, 2018). For CMIP6, CORDEX has yet to offer these firm recommendations. Instead, it contends that though model subselection approaches exist, there is no commonly accepted methodology on how to select a subset of GCMs for downscaling (CORDEX, 2021).

The questions are then: how should one select the representative CMIP6 subset for a specific task? How many simulations are necessary? Should those simulations come from independent models so that model agreement means something (Sanderson et al., 2015)? Should they come from models that are considered well suited in reproducing observed climate in a particular region or season to inspire fidelity in the projected outcomes (Ashfaq et al., 2022)? Should the subset prioritize having extreme cool/wet and hot/dry representatives, while also sampling possible climatic states in between (Qian et al., 2021)?





We posit that all three considerations, model individuality (henceforth, independence), model suitability for a task (henceforth, performance), and model outcome range (henceforth, spread), should be taken into account when subselecting from the CMIP archive. Existing subselection methods are typically based on two of the three considerations and can be broadly grouped into performance-based or spread-based categories.

While subselection can be based on performance alone (Ashfaq et al., 2022), studies that evaluate performance-based subselection tend to do so in conjunction with independence (Evans et al., 2013; Sanderson et al., 2015; Herger et al., 2018; Di Virgilio et al., 2022; Palmer et al., 2022). Evans et al. (2013) succinctly demonstrated that for small subsets to reflect the spread of larger ensembles, it is more important to account for model independence (defined in the study following Bishop and Abramowitz (2013)) than for model performance. Selection by model performance is usually anticipated to reduce ensemble spread, which can also pose issues if there is an interest in reproducing the mean of the base ensemble. Herger et al. (2018) established that an ensemble selected based on a performance ranking was sometimes worse at reproducing the base ensemble mean than an ensemble selected at random. Using a comprehensive method to select diverse and skillful model subsets from CMIP5, Sanderson et al. (2015) found the multi-model ensemble to be a "rather heterogeneous, clustered distribution, with families of closely related models lying close together but with significant voids in-between model clusters" via EOF analysis. CMIP5's interdependencies allowed for stages of subselection, first removing redundant simulations (without reducing the effective number of models), then removing poor performing simulations to improve ensemble mean mean state representation. More recently, Di Virgilio et al. (2022) and Palmer et al. (2022) built on these CMIP5-era strategies to support CMIP6 model subselection for CORDEX-Australasia and Europe, respectively. In Di Virgilio et al. (2022), CMIP6 models, represented by an individual ensemble member, were first filtered by performance for Australian climate applications, with top and mid-tier performers further evaluated for dependencies based on the methods of Bishop and Abramowitz (2013) and Herger et al. (2018). The study then went a step further to also assess climate change signal diversity to determine whether their high performing, independent subset effectively sampled the range of Australian climatic changes in CMIP6. In Palmer et al. (2022), a process-based European performance assessment for CMIP6 is presented. The study, an extension the work of McSweeney et al. (2015), also incorporates a second filter based on ClimWIP's dependence definition (Brunner et al., 2020b) and notably finds that regional model selection can differ from approaches targeting global metrics such as ECS that were central to CMIP5-era EURO-CORDEX recommendations.

Spread-based subselection or selection with the goal of maximizing climate change signal diversity, is often carried out either alone (e.g. Semenov and Stratonovich, 2015; McSweeney and Jones, 2016; Ruane and McDermid, 2017; Qian et al., 2021), or in conjunction with performance (Lutz et al., 2016) or independence (Mendlik and Gobiet, 2016). The clear application for this approach are impact studies where worst-case scenarios are often of interest. A common thread in spread-maximizing subselection studies is the concept of a "climate envelope", typically defined by changes in spatio-temporal aggregations of surface air temperature (SAT) and precipitation (PR) fields. For example, Lutz et al. (2016) selected models from a base ensemble initially based on projected changes in SAT and PR means, then refined the selection using changes and historical performance of SAT and PR extreme indices. Similarly, the Representative Temperature and Precipitation GCM Subsetting (T&P) approach, developed by Ruane and McDermid (2017), sampled SAT and PR changes in terms of deviation from their



respective ensemble medians. This allows for selected model combinations that span the cool/hot, wet/dry quadrants, as well as the "neutral" center, of the model ensemble. Qian et al. (2021) further advanced spread-maximizing subselection by evaluating the T&P approach against the Katsavounidis–Kuo–Zhang (KKZ) algorithm (Katsavounidis et al., 1994), in which members

are recursively selected to best span the spread of an ensemble. While both approaches had merit, the KKZ approach was more likely than the T&P approach to perform better than a randomly selected five-GCM subset in terms of both error in relation to the full-ensemble mean and coverage of the full-ensemble spread.

Despite the numerous model subselection approaches available, the process remains somewhat burdensome to users and often requires several rounds of iterative filtering before a subset of a user's desired size is reached. And challenges can emerge

depending on the choice of the starting filter: if performance is used as the starting filter, there is a risk the user is left with a set of very similar models that, though high performing, are not independent and perhaps do not effectively sample ensemble spread. If spread is used as the starting filter, there is no way for a user to ensure that the models they select projecting the worst-case scenarios are realistic to begin with. If independence is used as a starting filter, which is not a common practice but perhaps should be, the user can be assured that model agreement is equivalent to robustness, but may struggle to select the

highest performing or most unique projection from each model family.

To address these difficulties, we present an alternative approach to subselection that allows a user to simultaneously balance independence, performance, and spread interests and generate a subset of CMIP models of any size tailored to their specific application. The subselection method, Climate model Selection by Independence, Performance, and Spread (ClimSIPS; Merrifield and Könz, 2023), leverages a three-term cost function that grants the user freedom to decide how important independence,

performance, and spread are (relative to one another) for the application. For those concerned with the contribution of recognized model biases to downstream uncertainty, cost function parameters can be set to prioritize model performance. For those concerned with both performance and robust model agreement, the cost function can apportion 50% weight to its performance term and 50% to its independence term. And for those most concerned with sampling worst-case climate change outcomes, spread in mean state SAT and PR change space can be the primary consideration of the cost function, receiving perhaps 70%

of the total weight, while the remaining 30% is split between performance and independence.

We demonstrate ClimSIPS for European climate applications in the second part of this study. First, the remaining methodological inputs are defined, including a performance score (also derived from ClimWIP) based on climatological biases that affect projections of European climate and a multivariate SAT and PR change spread metric. We then discuss the mechanics of subselection: the independence, performance, and spread cost function minimization and its visual representation, the subse-

190 lection triangle. Because the cost function balances three interests, different combinations of models are selected as priorities shift. The subselection triangle, a ternary contour plot, summarizes which combination of models is optimal for each set of priorities.

ClimSIPS is demonstrated primarily within the CMIP6 ensemble for Central European summer climate applications, beginning with a five-model toy example. Upon extending the method to the full CMIP6 ensemble, we generate three model subsets

and formulate recommendations to help users navigate the subselection triangle. We compare ClimSIPS outcomes based on how a model is represented, whether by its ensemble mean (where applicable) or by an individual, spread-maximizing mem-



ber. Finally, we generate five model subsets for both Central European summer climate and Northern European winter climate applications in CMIP5/6.

Part II of this study is a case study of ClimSIPS for European climate applications, detailed in Section 5. Subsection 5.1 centers the definitions of performance and Subsection 5.2 the definitions of spread for European climate applications in the ClimSIPS protocol. The protocol is described in detail in Subsection 5.3 and resulting three and five model subsets for each combination of independence, performance, and spread prioritization are presented in Subsection 5.4. To close, concluding remarks are made in Section 6.

## 2 CMIP models

We begin our assessment with ensembles comprised of all models (and all initial condition/perturbed physics ensemble members therein) with historical simulations and the highest emissions projection pathways: Shared Socioeconomic Pathway 585 (SSP5-8.5) for CMIP6 model projections (Eyring et al., 2016; O'Neill et al., 2016) and Representative Concentration Pathway 8.5 (RCP8.5) for CMIP5 model projections (Taylor et al., 2012). For inclusion in Part I, the models also must provide (1) an estimate of ECS, calculated from a $4\times CO_2$ run using the Gregory method (Gregory et al., 2004) and (2) the following output fields (with their abbreviation and model output variable name given in brackets): near-surface 2-meter air temperature [SAT; tas], precipitation [PR; pr], and sea level pressure [SLP; psl]. Further inclusion into Part II's European case studies require the additional output fields of sea surface temperature [SST; tos], and all sky and clear sky downwelling shortwave radiation at the surface [rsds and rsdscs, respectively]. All fields are conservatively remapped onto a $2.5° \times 2.5°$ latitude–longitude grid. At the time of writing, 218 CMIP6 and 75 CMIP5 simulations met the aforementioned criteria for Part I and 197 CMIP6 and 68 CMIP5 simulations met the further criteria for Part II; additional CMIP6 simulations will be considered in subsequent publications as fields become available in the CMIP6 next generation archive, a standardized repository used by researchers at ETH Zurich (Brunner et al., 2020a).

The inclusion requirements each serve a specific purpose in the study. Historical SAT, SLP, and PR fields are explored as a means to set degrees of model dependence within the CMIP ensembles. The degrees of model dependence are then used to constrain ECS values through subsetting. Remaining historical model output fields establish model performance and SSP5-8.5/RCP8.5 projections establish mid-century climate change spread for Part II's European case studies.

Tables 1 and 2 provide a summary of the CMIP6 and CMIP5 models included in the study, respectively. We assign each uniquely named model (37 in CMIP6 and 29 in CMIP5) a numerical identifier (column 1) to be used throughout Part I. Model name and member count are also noted, with members labeled as initial condition ensemble members (IC), perturbed physics ensemble members (PP), or differently initialized ensemble members (DI) for multi-member ensembles. We provide additional information about members used in Supplementary Tables S1-S3, including full "ripf" identifiers for CMIP6 and "rip" identifiers for CMIP5. The IC designation corresponds to the "r" or realization index, the DI to the "i" or initialization index, and the PP to the "p" or physics index. The "f" or forcing index, unique to CMIP6, is shared by all members of each model.





**Table 1.** Summary of the CMIP6 Multi-Model Ensemble. Starred models meet the inclusion criteria for Part I only at the time of writing.

| ID | Model Name | Members | Family | ID | Model Name | Members | Family |
|---|---|---|---|---|---|---|---|
| 1) | ACCESS-ESM1-5 | 10 (IC) | SME | 20) | MPI-ESM1-2-HR | 2 (IC) | FAM |
| 2) | HadGEM3-GC31-MM | 4 (IC) | FAM | 21) | GFDL-CM4 | 1 | FAM |
| 3) | KACE-1-0-G | 3 (IC) | FAM | 22) | GFDL-ESM4 | 1 | FAM |
| 4) | ACCESS-CM2 | 3 (IC) | FAM | 23) | EC-Earth3* | 8 (IC) | FAM |
| 5) | HadGEM3-GC31-LL | 4 (IC) | FAM | 24) | EC-Earth3-Veg* | 4 (IC) | FAM |
| 6) | UKESM1-0-LL | 5 (IC) | FAM | 25) | FGOALS-f3-L | 1 | INDV |
| 7) | TaiESM1 | 1 | FAM | 26) | FGOALS-g3 | 4 (IC) | SME |
| 8) | CMCC-ESM2 | 1 | FAM | 27) | INM-CM4-8 | 1 | FAM |
| 9) | CMCC-CM2-SR5 | 1 | FAM | 28) | INM-CM5-0 | 1 | FAM |
| 10) | NorESM2-MM | 1 | FAM | 29) | MIROC6 | 50 (IC) | SME |
| 11) | CESM2-WACCM | 3 (IC) | FAM | 30) | MIROC-ES2L | 10 (IC) | SME |
| 12) | CESM2 | 5 (IC) | FAM | 31) | MRI-ESM2-0 | 2 (DI) | SME |
| 13) | CNRM-CM6-1-HR | 1 | FAM | 32) | E3SM-1-1 | 1 | INDV |
| 14) | CNRM-ESM2-1 | 5 (IC) | FAM | 33) | CanESM5 | 50 (IC,PP) | SME |
| 15) | IPSL-CM6A-LR | 6 (IC) | FAM | 34) | CAS-ESM2-0 | 2 (IC) | SME |
| 16) | CNRM-CM6-1 | 6 (IC) | FAM | 35) | GISS-E2-1-G | 6 (IC,PP) | SME |
| 17) | AWI-CM-1-1-MR | 1 | FAM | 36) | MCM-UA-1-0* | 1 | INDV |
| 18) | NESM3 | 2 (IC) | FAM | 37) | KIOST-ESM | 1 | INDV |
| 19) | MPI-ESM1-2-LR | 10 (IC) | FAM | **Totals (Members, Groups)** | | **218** | **19** |

Finally, to familiarize the reader with the concept of model families we will subsequently define, we also list the family group status of each model. The designation, "INDV", indicates a model is considered to be an individual represented by a single member. "SME" signifies that a model is represented by multiple members (e.g., initial condition ensembles, perturbed physics ensembles, combinations thereof) but is not determined to be part of a broader multi-model family. The "FAM" designation indicates a model is a member of a broader, multi-model family. In total, the 218 CMIP6 simulations from 37 uniquely named

models considered in Part I fall into 19 Groups (7 multi-model ensembles, 8 single model ensembles, and 4 individuals) and the 75 CMIP5 simulations from 29 uniquely named models fall into 20 Groups (8 multi-model ensembles, 5 single model ensembles, and 7 individuals). In Part II, 197 CMIP6 simulations from 34 uniquely named models and 68 CMIP5 simulations from 26 uniquely named models remain for the subselection exercise (Sup. Tabs. S1-S2).

## 3   Revisiting Model Dependence

In prior studies, it has been shown that a climate model's origins and evolution can be traced via statistical properties of its outputs (e.g. Masson and Knutti, 2011; Bishop and Abramowitz, 2013; Knutti et al., 2013). This indirect approach can uncover




**Table 2.** Summary of the CMIP5 Multi-Model Ensemble. Starred models meet the inclusion criteria for Part I only at the time of writing.

| ID | Model Name | Members | Family | ID | Model Name | Members | Family |
|----|-----------|---------|--------|----|-----------|---------|--------|
| 1) | ACCESS1-0 | 1 | FAM | 16) | GFDL-ESM2M | 1 | FAM |
| 2) | ACCESS1-3 | 1 | FAM | 17) | GFDL-CM3 | 1 | INDV |
| 3) | HadGEM2-ES | 4 (IC) | FAM | 18) | MIROC5 | 3 (IC) | SME |
| 4) | NorESM1-ME | 1 | FAM | 19) | MIROC-ESM | 1 | INDV |
| 5) | NorESM1-M | 1 | FAM | 20) | GISS-E2-H | 5 (IC,PP) | FAM |
| 6) | CCSM4 | 6 (IC) | SME | 21) | GISS-E2-R | 5 (IC,PP) | FAM |
| 7) | CESM1-CAM5 | 3 (IC) | SME | 22) | bcc-csm1-1 | 1 | FAM |
| 8) | IPSL-CM5B-LR | 1 | INDV | 23) | bcc-csm1-1-m | 1 | INDV |
| 9) | IPSL-CM5A-MR | 1 | FAM | 24) | BNU-ESM* | 1 | FAM |
| 10) | IPSL-CM5A-LR | 1 | FAM | 25) | inmcm4 | 1 | INDV |
| 11) | EC-EARTH* | 5 (IC) | FAM | 26) | CanESM2 | 5 (IC) | SME |
| 12) | CNRM-CM5 | 5 (IC) | FAM | 27) | MRI-CGCM3 | 1 | INDV |
| 13) | MPI-ESM-MR | 1 | FAM | 28) | CSIRO-Mk3-6-0 | 10 (IC) | SME |
| 14) | MPI-ESM-LR | 3 (IC) | FAM | 29) | FGOALS-g2* | 1 | INDV |
| 15) | GFDL-ESM2G | 1 | FAM | **Totals (Members, Groups)** | | **75** | **20** |

hidden dependencies within the ensemble, e.g. models that are similar because they share components or lineages, but not names. It also has the advantage that it does not presume model similarity based on name alone; models in active development can evolve substantially from version to version (e.g. Boucher et al., 2020; Danabasoglu et al., 2020) while models from

different modeling centers can be quite similar. For these reasons, we employ this output field similarity strategy, updated from the current version of the ClimWIP independence weighting scheme (Brunner et al., 2020b), to revisit the concept of model families within CMIP.

  The ClimWIP independence weighting scheme defines model dependence using an intermember distance metric based on long-term, large-scale climatological averages (Merrifield et al., 2020). The rationale behind this underlying spatio-temporal

aggregation is that it is able to identify an initial condition or perturbed physics ensemble as a single model (by averaging over differences due to internal variability or parameter uncertainty) while simultaneously maintaining varying degrees of differentiation between models in the ensemble overall. In practice, this balance between reducing intra-model or "within-model" intermember spread while still preserving inter-model or "between-model" intermember spread is key to a useful definition of dependence within CMIP. It was found that the absolute values of global-scale annual average SAT and SLP

climatologies are able to achieve this balance (Merrifield et al., 2020), but to what extent has not yet been evaluated.

  Here we explicitly investigate the within-model vs. between-model spread balance in ClimWIP's independence predictors to ensure they provide a suitable application-agnostic definition of model dependence for atmospheric studies. This is done by testing the sensitivity of the final root-mean-square error (RMSE) intermember distance metric to each methodological choice





in ClimWIP, including temporal averaging period, spatial masking strategies, and predictor field choices. Intermember distance
is calculated through pairwise RMSE between ensemble members for each predictor field individually, then predictor RMSEs
are normalized by their respective ensemble mean values and averaged together to obtain a single distance for each member
pair. To first order, the intermember distance metric is robust to methodological choices; the sensitivity testing did not reveal
major shifts in whether a model was considered relatively dependent or independent with respect to the other models in the
ensemble. However, refining each methodological choice sharpens dependence delineations along the spectrum of dependence
and lends further credence to the concept of model families.

The first methodological choice we revisit is the length of the climatological period of the global SAT and SLP predictors
(Figure 1). To reduce internal variability on decadal timescales, we extend the predictor climatological period from 1980-2014
(Brunner et al., 2020b) to 1905-2005, a common 101 years from the historical period of both CMIP5 and CMIP6. Illustrat-
ing the effect in the CMIP6 ensemble, we find reduced intermember distances between initial condition ensemble members,
highlighted in color, for the 1905-2005 averaging period (Fig.1a) compared to the 1980-2014 period (Fig.1b). This grouping
of known dependencies helps to further distinguish initial condition / perturbed physics ensemble members from members of
other models (Fig.1, light gray). This is particularly clear in the case of CESM2-WAACM, the longer climatological averaging
period allows its three ensemble members to be distinguished from those of CESM2; with the shorter period, the two CESM2
model variants overlap (Fig.1b).

As the CMIP6 historical record spans 1850-2014 (Eyring et al., 2016) and the CMIP5 historical record spans 1870-2005
(Taylor et al., 2012), our choice of a 101-year averaging period could have been extended further back in time. However, we
find that increasing the period back into the 19th century does not appreciably change intermember distances. Additionally,
the 1905 start date allows for further assessment, should a future user be inclined, of how the independence predictor fields
compare to observed fields. Some observed products, such as the global Berkeley Earth Surface Temperature product, have
reduced spatial coverage in the 19th-century (Rohde et al., 2013).

The second methodological choice of interest is whether the dependence definition benefits from a spatial mask applied
to the global SAT and SLP predictors. Spatial masking may not be a necessity; within-model spread can be reduced through
temporal averaging, as seen in Fig. 1 and some level of between-model spread is provided by the choice to use predictor
absolute values (Merrifield et al., 2020). Predictor absolute values provide between-model spread because it has not been a
priority, historically, to calibrate or tune a model towards the absolute value of observed SAT or SLP. Absolute biases with
respect to observations (estimates themselves) tend to be seen as less important metrics of model performance than relative
change with respect to a historical base period (Mauritsen et al., 2012; Hourdin et al., 2017). The absolute value of global SAT
in particular has been identified as an emergent property of climate models, reflecting differences underpinned by different
model components and physical parameterizations (Schmidt, 2014). It is conceivable that in the future, however, the reduction
of absolute global biases will become more of a priority to modeling centers and the between-model spread we use to determine
model diversity will disappear. Several emergent properties defined in the CMIP5-era have vanished in CMIP6, making this a
credible concern (Simpson et al., 2021; Sanderson et al., 2021).



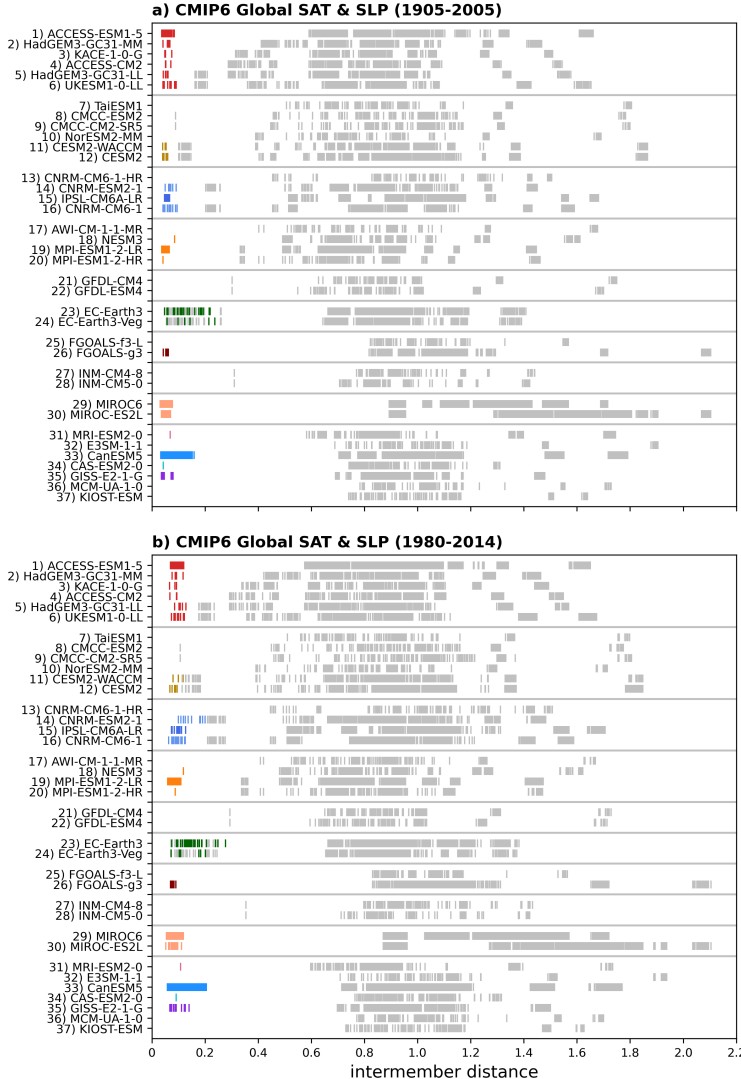

**Figure 1.** Intermember distances in CMIP6 based on Global SAT and SLP climatological fields averaged over the period (a) 1905-2005 and (b) 1980-2014. For each model, distances between initial condition or perturbed physics ensemble members are marked in color and distances to members of the remaining models are marked in light gray.

Spatial masking can help guard against independence predictor convergence because an atypically-masked model output field is unlikely to feature in traditional model evaluation exercises. Further, fingerprints, created through spatial masking, can

be explicitly designed to achieve our dependence objectives. We design a spatial fingerprint, shown in Figure 2 for CMIP6 and Supplementary Figure S1 for CMIP5, that bolsters between-model spread and reduces within-model spread in the ClimWIP independence predictor fields. The SAT and SLP fingerprints, shown superimposed on their ensemble mean annual average





climatologies (1905-2005) in Figure 2e,f for CMIP6 and Supplementary Figure S1e,f for CMIP5, define model dependence for the remainder of the study.

The fingerprint design is conceptually simple; between-model spread is amplified by masking regions where it is low (Fig.2, square hatching) and within-model spread is damped by masking regions where it is high (Fig.2, diamond hatching). Though between-model spread is difficult to clearly define within CMIP's multi-member, multi-model structure, it can be estimated via standard deviation across an ensemble comprised of one ensemble member per model. The first member is selected from each multi-member ensemble: r1i1p1 in CMIP5 and r1i1p1f1 where available in CMIP6, with exceptions listed in Supplementary

Table S4. Upon computing the standard deviation across the one member per model ensembles, we mask out the region where between-model spread is at or below its 15th percentile (Fig.2a,b, square hatching). This "low" between-model spread is largely confined to subtropical oceanic regions for both the SAT and SLP 1905-2005 climatologies.

In addition to regions of low between-model spread, we also select and mask regions of high (at or above the 85th percentile) within-model spread (Fig.2c,d, diamond hatching). CMIP6 within-model spread is represented in Fig.2c,d by the

median of the standard deviations within the 12 CMIP6 initial condition ensembles with five or more members (ACCESS-ESM1-5, CanESM5, CESM2, CNRM-CM6-1, CNRM-ESM2-1, EC-Earth3, GISS-E2-1-G, IPSL-CM6A-LR, MIROC-ES2L, MIROC6, MPI-ESM1-2-LR, and UKESM1-0-LL). CMIP5 within-model spread is similarly defined within five initial condition ensembles (CanESM2, CCSM4, CNRM-CM5, CSIRO-Mk3-6-0, and EC-EARTH). Because the five or more member requirement necessitates that we use a set of models to define internal variability rather than the full ensemble, we evaluate

within-model spread within each individual model ensemble in Supplementary Figures S2 and S3 for SAT and SLP climatology, respectively. For SAT climatology, most models share regions of elevated internal variability across the Arctic and in particular, in the vicinity of the annual climatological sea ice edge in the Irminger and Barents seas (Fig.2c; Davy and Outten, 2020). For SLP climatology (Fig.2d), internal variability remains in parts of the Arctic and Antarctic, masking the Antarctic Polar high region where between-model variability is also at a maximum (Fig.2b). Because patterns of elevated internal

variability are broadly similar among the models evaluated, we make the assumption that this within-model spread estimate is transferable to the other models in the ensemble that lack additional initial condition ensemble members. Masking regions where high within-model spread and high between-model spread coincide eliminates the possibility that the between-model spread is actually internal variability in disguise, possibly due to the presence of very similar but differently named models in the one-member-per-model ensemble.

It is important to note that results are not highly sensitive to precise percentile thresholds used to define low between-model spread and high within-model spread; intermember distances are largely consistent for thresholds between the 5th and 20th percentile for between-model spread and the 80th and 95th percentile for within-model spread. The 15th and 85th percentiles were chosen to limit the percentage of masked grid points to no more than 30% of the domain total, similar in extent to a land mask. Masking the majority of the points in the domain increases the risk of relying on small-scale biases to

define dependence, which complicates the interpretation of models being dependent because they are spatially similar overall. Masking very few points does not refine intermember distances much beyond those based on unmasked predictors (as used in Fig. 1) thus rendering the exercise unwarranted.



The third and final methodological choice we investigate is that of the fields in ClimWIP's independence predictor set. Due to the complexity and breadth of model output, innumerable combinations of different climatic fields can be put forth to

define dependence. Because we aim for a dependence definition that is broadly applicable to studies of surface climate, we also considered PR as an addition to the independence predictor base set. However, we found that the inclusion of PR did not promote our primary goals: to group known dependencies and differentiate between models. The spatially masked annual-average PR climatology predictor, shown in Supplementary Figure S4, tended to reduce between-model differentiation within the ensemble as a whole, likely due to the majority of its between-model spread being co-located with high within-model

spread in the tropical rain belts associated with the Intertropical Convergence Zone (ITCZ). For this reason, we chose to move forward with a dependence definition based solely on SAT and SLP fingerprints.

## 4    Model families and their influence on CMIP uncertainty

Refining ClimWIP's dependence definition aids our effort to define model families within CMIP5/6. We pursue defining model families because many downstream applications, including ClimSIPS, benefit from a discrete definition of dependence rather

than a continuous dependence spectrum. To achieve the discrete definition of dependence, each CMIP5/6 model is designated as either a single model ensemble, part of a model family, or an individual (see Tables 1 and 2) based on intermember distances within the ensemble. We then make an effort to verify the designations through published model descriptions and reported metadata.

In Figure 3, we show how intermember distances based on the sum of normalized RMSEs calculated from SAT and SLP

fingerprints help to uncover model relationships within CMIP. Intermember distances are presented for each model in one dimension (Fig.3a,c) and, as recommended by Abramowitz et al. (2019), for the ensemble as a whole in a low dimensional projected space (Fig.3b,d). The low-dimensional projection is obtained through a standard metric multidimensional scaling (MDS) approach. The MDS method embeds the N-dimensional CMIP distance matrices into two-dimensional space while attempting to preserve relative positioning between models (Borg and Groenen, 2005). To assist the MDS method with model

positioning, we ensure that ensemble members from each model are initially placed together and can thus settle into their final positions as a group. Without this initialization, there is a risk that an ensemble member may get stranded away from its group as the method contends with how best to map N-dimensions to two dimensions.

In both one and two-dimensional visual representations, it is clear that the ensemble of opportunity has grown from CMIP5 (Fig.3, bottom panels) to CMIP6 (Fig.3, top panels); there are more uniquely named models in CMIP6 than in CMIP5 and

on average, more ensemble members per model. In projected space (Fig.3b,d), models with multiple ensemble members are highlighted using a "radius of similarity" (shaded circles), a construct also conceived by Abramowitz et al. (2019). Here we employ this construct as largely a visual aid and set the radius to 2.5 times the maximum deviation of an individual ensemble member from its ensemble mean. Models are labelled by number in the projections with numbers listed in Tables 1 and 2 and on the y-axis of Fig.3a,c.





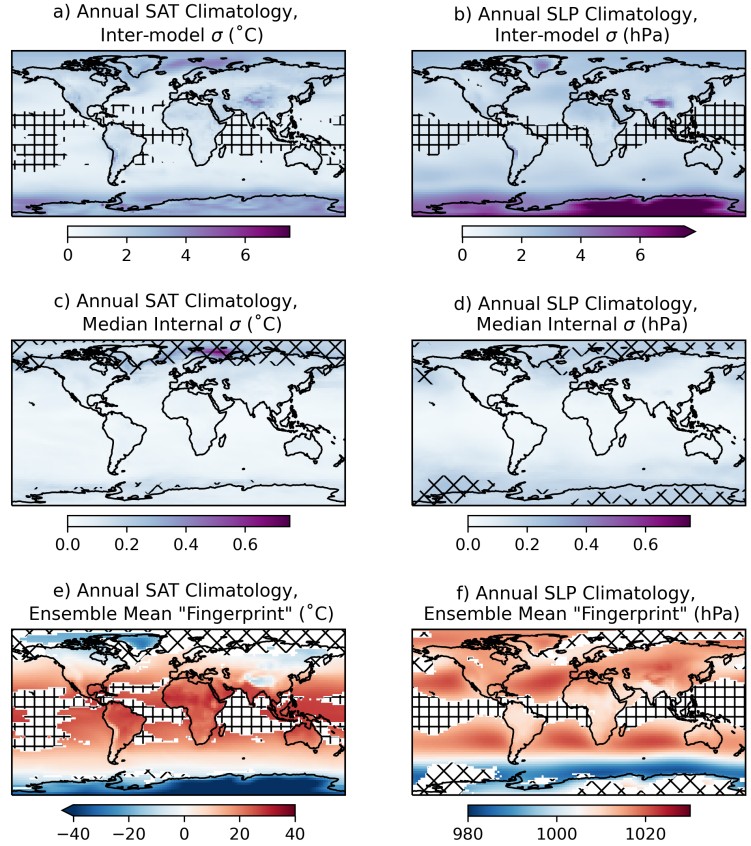

**Figure 2.** Determining the spatial "fingerprint" within the fields used to identify CMIP6 climate model dependence: annual mean SAT (°C) and SLP (hPa) climatology averaged over the period 1905-2005. (a,b) a measure of between-model spread of the dependence predictors computed as the standard deviation ($\sigma$) across a CMIP6 ensemble with only one member per model (see Sup. Table S4). Square hatching indicates where between-model spread is low, at or below its 15th percentile (calculated based on the spatial field). (c,d) Median internal variability of the dependence predictors computed as the median of the standard deviations within the 12 CMIP6 initial condition ensembles with five or more members (ACCESS-ESM1-5, CanESM5, CESM2, CNRM-CM6-1, CNRM-ESM2-1, EC-Earth3, GISS-E2-1-G, IPSL-CM6A-LR, MIROC-ES2L, MIROC6, MPI-ESM1-2-LR, and UKESM1-0-LL). Diamond hatching indicates where median internal variability is high, at or above its 85th percentile. (e,f) Fingerprint used to determine dependence, shown as the ensemble mean climatology of the whole CMIP6 ensemble with the regions of low between-model spread and high internal variability masked and hatched with square and diamond hatching respectively.

In CMIP6, an ensemble "core" comprised of all but two models has emerged; the intermember distance metric identifies MIROC6 and MIROC-ES2L as considerably more independent from the rest of the ensemble (Fig.3b). We use a broken axis in CMIP6's low dimensional space projection to accommodate the two MIROC outliers and emphasize the structure of the



CMIP6 core. In contrast, CMIP5 does not have the same level of core and outlier structure, and intermember distances create a more distributed dependence spectrum (Fig.3d) similar to the one described in Sanderson et al. (2015).

In the one-dimensional representation, distances between a model's ensemble members are shown in color, distances to family members are shown in dark gray, and distances to the rest of the ensemble are shown in light gray (Fig.3a,c). Beginning with the most dependent entities in CMIP, the SAT and SLP fingerprint metric clusters initial condition ensemble members at distances of around 0.05 in all but one case. The exception, EC-Earth3 and EC-Earth3-Veg (Fig.3a, dark green, 23 and 24), exhibit overlapping intermember distances from 0.08 to 0.20 due to remaining decadal variability in the predictors traceable

to oscillations in the EC-Earth3 piControl run from which ensemble members of both models are branched (Döscher et al., 2022). At the next level of dependence, the intermember distance metric introduces a measure of disambiguation between initial condition and perturbed physics ensemble members, as illustrated by two models in CMIP6, CanESM5 (Fig.3a, bright blue, 33) and GISS-E2-1-G (Fig.3a, bright purple, 35). Strikingly, in Fig. 3b, CanESM5's two 25-member initial condition ensembles can be seen clearly as two distinct clusters in two-dimensional space. CanESM5 initial condition ensembles are

reported to differ by wind stress remapping; conservative remapping is used for "p1" members, and bilinear regridding is used for "p2" members (Swart et al., 2019).

Continuing along the spectrum of dependence from most dependent to most independent, intermember distances reveal model similarities that would require high-level knowledge of CMIP model origins to determine *a priori* (Fig.3a,c dark gray). In this regime, where models are separated by distances of around 0.1 to 0.6, subjective decisions must be made regarding

whether or not a model is part of a family. We chose two criteria to determine if a family should be formed: (1) a model family must be a self-contained group, i.e. all family members must be closer to each other than to other models, and (2) models within the family must have a median intermember distance to the rest of the family that is less than 0.56. This median intermember distance threshold was based specifically on the composition of CMIP6 to ensure that we did not simply define one large family within the ensemble's core (Fig.3b). However, because it is ultimately a subjective threshold, we pursued further justification

of model families in the literature.

To ensure that similar models form self-contained groups, we match intermember distances between pairs of models in one-dimensional space. For example, CMIP6's INM-CM4-8 and INM-CM5-0 are separated by a distance of 0.32 from each other as indicated by a dark gray line in their respective rows in Fig. 3a. To assist with model pair matching, we ordered and used mutual colors for models that we anticipated would be similar enough to be grouped into families. In general, we predicted that

models contributed by the same modeling center might be family members, then set about to determine if the assumption was substantiated by intermember distances. We also anticipated three "extended" families based on an analysis of model metadata, summarized in Sup. Tabs. S1 and S2. The first, shown in dark red (models 1-6) in Fig.3, is comprised of models with UK Met Office Hadley Centre atmospheric components. In CMIP6, intermember distances show five of the six models highlighted in red on the y-axis of Fig.3a, satisfy both the self-contained group and median intermember distance threshold criteria to form

a family. This grouping makes sense as all five models (HadGEM3-GC31-MM, KACE-1-0-G, ACCESS-CM2, HadGEM3-GC31-LL, and UKESM1-0-LL) use the same MetUM-HadGEM3-GA7.1 atmospheric component (Sup. Table S1). The sixth model, ACCESS-ESM1-5, is closer to other models in CMIP6 than it is to its anticipated family members, likely because it





**Figure 3.** Intermember distances used to identify degrees of dependence within (a) CMIP6 and (c) CMIP5. For each model, within-model distances (i.e., initial condition ensemble members or perturbed physics ensemble members) are marked in color, distances to members of other similar models are marked in dark gray and distances to members of the remaining models are marked in light gray. Models grouped into families are highlighted on the y-axis. To better visualize levels of similarity within the multi-model ensembles, CMIP6 (b) and CMIP5 (d) intermember distances are projected from high dimensional space into two dimensions using multidimensional scaling. Models are colored and labelled numerically as indicated in panels a and c. Initial condition and perturbed physics ensembles are given a radius of similarity (shaded circles) equivalent to 2.5 times the maximum deviation from their ensemble mean. Note that in panel b, a broken axis is used to emphasize the structure of the primary CMIP6 model cluster with respect to two independent constituents, MIROC6 and MIROC-ESL.





reports to use a CMIP5-era HadGAM2-based atmospheric component rather than a CMIP6-era MetUM-HadGEM3-GA7.1 atmospheric component. In CMIP5, a similar family of models with UK Met Office Hadley Centre atmospheric components is

present (Fig.3c, dark red, models 1-3), where it is comprised of three uniquely named models, ACCESS1-0, ACCESS1-3, and HadGEM2-ES. ACCESS1-0 and HadGEM2-ES also share HadGAM2 atmospheres, while ACCESS1-3 features a modified version of the UK Met Office Global Atmosphere 1.0 AGCM (UM7.3/GA1; Bi et al., 2012; Brands, 2022). Despite the differing atmospheric component, ACCESS1-3 is closer to ACCESS1-0 and HadGEM2-ES than to other CMIP5 models and thus joins the family group.

The second anticipated extended family, shown in gold (models 7-12), features models with atmospheres that share commonalities with the National Center for Atmospheric Research (NCAR) Community Atmosphere Model (CAM). In CMIP6, there is a gap in pairwise intermember distance between models with a CAM5.3 atmosphere (CMCC-ESM2, CMCC-CM2-SR5) and models with a CAM6 atmosphere (CESM2 and CESM2-WAACM). Two additional models, TaiESM1 and NorESM2-MM, are similar enough to also be included in the family (Fig.3a, gold highlight) likely because their atmospheres are based on CAM5.3

and CAM6, respectively, with several alternative parameterizations incorporated (Lee et al., 2020; Seland et al., 2020). Though NorESM2-MM is closer to the CAM6-based models than the CAM5.3-based models in terms of intermember distance, it does end up placed towards the CAM5.3-based cluster in low dimensional space due to how the MDS method chooses to optimize relative positioning (Fig.3b). In CMIP5, there is less similarity seen between members of the CAM-based anticipated extended family (Fig.3c, gold, models 4-7), particularly between CESM1-CAM5 and the models based on CAM4, its predecessor atmo-

spheric component (see Sup. Table S3). The four models (NorESM1-ME, NorESM1-M, CCSM4, and CESM1-CAM5) reside in the same region of low dimensional space, but do not form a discernible cluster (Fig.3d) and do not satisfy either criteria to be considered one extended family. Instead, NorESM1-ME and NorESM1-M form a family (Fig.3c gold highlight) while CCSM4 and CESM1-CAM5 remain as single model ensembles.

The third anticipated extended family, shown in orange (models 7-12), is made of models that utilize ECHAM6 atmospheric

components developed at the Max Planck Institute for Meteorology. In CMIP6, a gap is present between within- (Fig.3a color) and between-model distances (Fig.3a dark gray) in the grouping, which may be traceable to differences in horizontal resolution (Sup. Table S1). This anticipated family has also grown from CMIP5, which featured two ECHAM6.1-based model variants that differ by vertical atmospheric resolution and horizontal ocean resolution (Giorgetta et al., 2013), to CMIP6, which features four ECHAM6.3-based models contributed by different modeling centers. The family is positioned in a cluster towards the

center of both CMIP ensembles in low dimensional space (Fig.3b,d).

In addition to the three anticipated families, several other families emerge upon assessing intermember distances. In CMIP5, the EC-EARTH and CNRM-CM5 initial condition ensembles share a level of similarity on par with the other families, as do bcc-csm1-1 and BNU-ESM (Fig.3c). In CMIP6, we find the three CNRM models to be similar enough to IPSL-CM6A-LR to satisfy the family-criteria (Fig.3a light blue and medium blue). Similarity in these cases cannot be traced to a particular

atmospheric component model, but for CNRM and IPSL, similarity could have arisen through an effort to foster collaboration between the two French modeling groups after CMIP5 (Mignot and Bony, 2013). The remainder of model families in both CMIP5 and CMIP6 feature models originating from the same modeling center. However, not all same center models are similar





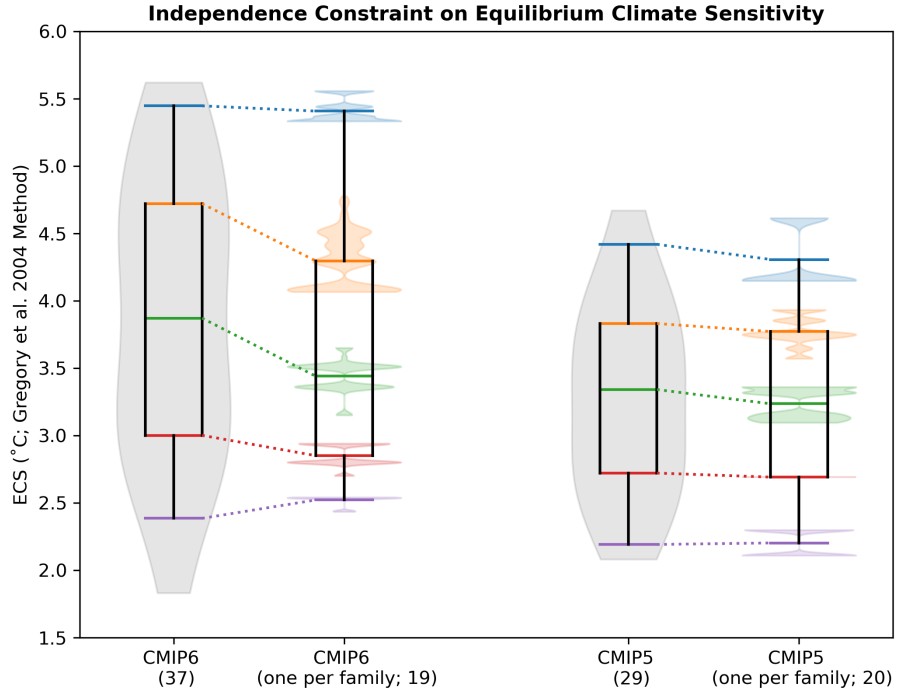

**Figure 4.** Comparison between the full distribution and the "one per family" subset distribution of effective equilibrium climate sensitivity (ECS) in CMIP6 and CMIP5. Full distributions are shown as a violin plot (gray) superimposed with the median (green) and the 5th (purple), 25th (red), 75th (orange), and 95th (blue) percentiles. One-per-family subset distributions of each percentile (violins) reflect 10,000 subsets from a bootstrap random selection of one model from each model family (see Figure 3). The means of each percentile distribution are used to create the one-per-family box and whisker. The number of members in each distribution are given in parentheses.

enough, in terms of intermember distance, to be considered potential relatives. For example, GFDL-CM3 is more similar to other CMIP5 models than it is to Earth system models from the same modeling group, GFDL-ESM2M and GFDL-ESM2G

(Fig.3d). In this case, a different atmospheric component model version accompanies the dissimilarity in historical model output; GFDL-CM3 uses a later generation atmospheric component than GFDL-ESM2M and GFDL-ESM2G (See Sup. Table 3). Meanwhile, GFDL-ESM2M and GFDL-ESM2G differ from each other only by ocean component (Dunne et al., 2012) and do satisfy the criteria to form a family. CMIP6's FGOALS-f3-L and FGOALS-g3 are also found to be relatively distinct from each other in terms of intermember distance; the two models differ in atmospheric component, notably by atmospheric

finite differencing method (Zheng et al., 2020). The only models to share an atmospheric component and not form a family are CMIP6's MIROC6 and MIROC-ES2L. Though the two MIROC variants form a self-contained group, they are more distinct from each other in terms of intermember distance than most models pairs considered to be independent within the CMIP6 core and are thus considered independent single model ensembles instead of a family.





One of the primary reasons we define model families is to enhance our understanding of how dependence influences CMIP
uncertainty estimates. Model families establish a stricter definition of independence within CMIP than the "one model, one
vote" standard typically employed in multi-model assessments if weights (fractional votes) are not desired or possible (Knutti,
2010). The one model, one vote standard treats all uniquely named models in the ensemble as independent and allows them
each to be represented by one simulation. By this standard, CMIP6 is represented by 37 independent entities and CMIP5
is represented by 29 independent entities. We compare this traditional approach against a "one family one vote" standard,
where each model family, single model ensemble, and individual is represented by one simulation. This reduces CMIP6's
representation to 19 and CMIP5's to 20 independent entities.

We assess the impact of the one family, one vote independence constraint on distributions of ECS, a key climate metric
reflecting the magnitude of warming a model projects in response to $CO_2$ doubling from preindustrial levels (Charney et al.,
1979). We source ECS values primarily from the IPCC (Smith et al., 2021), and when not available, from studies reporting to
compute it via the Gregory et al. (2004) method. Further information on the sourcing of ECS is provided in the supplement; the
ECS values used are shown in Supplementary Figure S5. Raw distributions of ECS in CMIP5/6 are represented in Figure 4 by
both violin (gray shading) and box and whisker elements. The violin representation gives a sense of how the shape of the ECS
distribution has evolved from CMIP5 to CMIP6, with CMIP6 having a more bimodal structure, a lighter low-ECS tail, and a
heavier high-ECS tail than CMIP5. This is consistent with the highly-publicized finding that a subset of CMIP6 models are
"running hotter" than their CMIP5 predecessors (e.g. Flynn and Mauritsen, 2020; Zelinka et al., 2020; Tokarska et al., 2020);
there are only five models with an ECS above $4°C$ in the CMIP5 distribution compared to 17 in the CMIP6 distribution. Box
and whisker elements, superimposed on the violins, provide a way to investigate how different percentiles of the distribution
compare between CMIP generations and shift under the new one family, one vote independence constraint. We focus on the 5th
(purple), 25th (red), median (green), 75th (orange), and 95th (blue) percentiles. All percentiles have increased between CMIP5
and CMIP6, ranging from the 5th, which increases by $0.19°C$ (2.19 to $2.38°C$), to the 95th, which increases by $1.03°C$ (4.41
to $5.45°C$). Also notable, no CMIP5 model has an ECS that exceeds CMIP6's 75th percentile of $4.72°C$ (see Sup. Fig. 5).

To ascertain if model dependence can explain the shift in ECS between CMIP generations, we apply the one family, one vote
independence constraint to ECS in both ensembles via a bootstrap protocol. First, base ensembles are formed from the models
(single model ensembles and individuals) already represented by one ECS value. Subsequently, one member of each family
is randomly selected, and its ECS value joins the base ensemble to form a "one per family" ensemble. Percentiles are then
computed, and the procedure is repeated 10,000 times to generate distributions of percentiles (Fig.4 color-coordinated violin
elements). Percentile distributions reflect that model families span a range of ECS values and the one-per-family distribution
shifts depending on the combination of models selected. Finally, the overall one-per-family ensemble box and whisker element
is constructed from the means of each percentile distribution.
In CMIP6, there are seven families, comprised of two to six models, to randomly select from (Fig.3a label highlights).
After 10,000 rounds of selection, the average CMIP6 one per family distribution (Fig.4 second element from left) has reduced
skewness towards high ECS compared to the raw CMIP6 distribution. The removal of dependent entities does not affect
CMIP6's 95th percentile (Fig.4 blue; $5.4°C$) due to the certainty that at least two of the 19 models in the one per family



distribution have an ECS above $5°C$ (E3SM-1-1 and CanESM5; Sup. Fig. 5). In contrast, the interquartile range (Fig.4 red to

orange) of CMIP6's one per family distribution is shifted toward lower values of ECS with respect to the raw distribution, to

$2.85 - 4.29°C$ from $3.0 - 4.72°C$. CMIP6 median ECS also shifts down by $0.43°C$ to $3.44°C$ when representation is limited

to one family, one vote. This suggests that the higher ECS mode of CMIP6's bimodal distribution is due, in part, to there being

more "copies" of higher ECS models in the ensemble. Removing redundancies also constrains the lower tail of the distribution

(Fig.4 purple), which is set in the raw ensemble by the two models with ECS below $2°C$, family members INM-CM4-8 and

INM-CM5-0.

In CMIP5, of eight families, seven are comprised of two models and one is comprised of three models (Fig.3c label high-

lights). Selecting from CMIP5's smaller families (compared to CMIP6) results in a CMIP5 one per family distribution that is

nearly identical to the raw CMIP5 distribution (Fig.4 right). Limiting family representation does have a marginal impact on

the CMIP5 95th percentile and median, shifting them each down by $0.11°C$, but does not skew the distribution nor narrow

its interquartile range as it does in CMIP6. This suggests the approach taken in IPCC AR5 where model dependence was not

explicitly considered was reasonable. While dependencies exist in CMIP5, they happen to be distributed in a way that the mean

and overall model spread is not strongly affected. We find that dependence alone cannot account for the full distributional shift

in ECS between CMIP5 and CMIP6, but does reconcile the two somewhat, reducing the difference for CMIP6 and CMIP5

median ECS by over $60\%$.

Ultimately, constraining by independence emphasizes that though there are significantly more simulations in CMIP6 than in

CMIP5 (here 218 versus 75), there are not significantly more independent models in CMIP6 as of yet. Highly similar models

appear more frequently in CMIP6 under different names, and increased representation has just happened to occur more for

model families on the high end of the ECS distribution. It is important to note that limiting representation in this instance is not

a comment on model quality in any way, it is only a comment on whether a model's historical output is sufficiently independent

of other models in the ensemble. Because of the influence redundancies have on multi-model uncertainty distributions, model

families are crucial for users to be aware of, whether or not they chose to sub-sample CMIP6.

## 5 ClimSIPS for European climate applications

For use cases that require a subset of CMIP models, model dependence is one of three common ensemble design consid-

erations. Equally important to subselection are model performance and spread between model outcomes in the chosen set.

Discussed in the following subsections, performance is defined with respect to observations over different periods of the his-

torical record. Spread is calculated from projected regional changes between present climate, averaged from 1995-2014, and

mid-century climate, averaged from 2041-2060 in CMIP6's SSP5-8.5 or CMIP5's RCP8.5 emissions scenario. Performance

and spread definitions were designed to select sets of models to underpin European regional climate modeling efforts and

impact assessments.





## 5.1 Performance Metric

Performance centers on properties of a model that make it suited to simulating future European climatic states as defined by a multivariate model-observation comparison metric. We aim to identify models with historical biases that would preclude them from accurately projecting future European climate rather than attempting to elevate one model over another based on its success in simulating a limited set of historical European climate variables. We focus on historical biases because all CMIP models have strengths and weaknesses in simulating aspects of the climate system and it is not always clear that model's historical strengths will translate into future skill (Weigel et al., 2010). Historical biases, in contrast, highlight cases where models lack important dynamic or thermodynamic processes (Knutti et al., 2017) or are simply too hot, cold, wet, or dry to transition into a realistic future temperature or precipitation regime (Eyring et al., 2019).

Specifically, we compare all CMIP members with observations using ClimWIP's performance weighting strategy (Brunner et al., 2020b). We utilize predictor fields relevant to two European case studies: Central European (CEU) summer (June-July-August; JJA) and Northern European (NEU) winter (December-January- February; DJF) SAT and PR change between 1995-2014 and 2041-2060 mean states. The two European regions assessed correspond to the CEU and NEU SREX regions used by the IPCC (Iturbide et al., 2020). Hereafter, we describe a mix of local, regional, and global climatological predictors, including a base set of four annual-average predictors used in both cases and two additional seasonal predictors specific to each case. The four predictor base set includes annual-average European SAT climatology over two base periods (1950-1969, 1995-2014), annual-average North Atlantic sea surface temperature (SST) climatology (1995-2014), and annual-average Southern Hemisphere midlatitude shortwave cloud radiative effect (SWCRE) climatology (2001-2018). We define SWCRE as the difference between all and clear sky downwelling shortwave radiation (rsds-rsdscs) at the surface (Cheruy et al., 2014). For the Central European summer case, additional relevant predictors include the JJA average climatologies of gridded Central Europe Station PR (1995-2014) and CEU SWCRE (2001-2018). For the Northern European winter case, DJF average climatologies of gridded Northern Europe Station PR (1995-2014) and North Atlantic Sector SLP (1950-2014) are used. Further details on predictor regions and masks are provided in Supplementary Section 4.

In both summer and winter, local predictors have the potential to reveal specific historical biases that erode confidence in future SAT and PR projections. For example, summer radiation biases (due to biases in local cloud cover) may affect a model's ability to warm a realistic amount in the future. Potentially persistent summer precipitation biases may also affect warming biases further through moisture availability and local land-atmosphere interaction issues (Fischer et al., 2007; Sippel et al., 2017; Ukkola et al., 2018). In winter, local precipitation biases, which are common at the grid resolution scales of GCMs, may signify a model's inability to represent processes relevant to precipitation change, such as ocean eddies and extratropical cyclone activity (Moreno-Chamarro et al., 2021).

On regional scales, predictors serve to indicate potential process-based simulation issues that may affect both past and future European climate. We employ two periods of annual-average European SAT climatology to establish whether a model's historical SAT response to aerosol emission-control measures in Europe is in line with observations (Haug et al., 2004). The use of two climatological periods also helps to avoid penalizing models for differing from observations by chance over a 20-year



period due to internal variability (Deser et al., 2012). Additionally, we include annual-average North Atlantic SST climatology
because SST biases in the region have been linked to biases in European SAT and PR variability through interactions with
atmospheric circulation (e.g. Keeley et al., 2012; Simpson et al., 2019; Borchert et al., 2019; Athanasiadis et al., 2022). As
atmospheric circulation biases tend to be more pronounced in the winter than in the summer, we also explicitly incorporate
mean state SLP in the North Atlantic sector in the winter predictor set. Mean state SLP serves as a potential indicator of biases
in the storm track and the frequency of prevailing weather regimes, both primary drivers of winter SAT and PR variability (e.g.
Simpson et al., 2020; Harvey et al., 2020; Dorrington et al., 2021).

Finally, with the advent of CMIP6 and models with high climate sensitivity, we incorporate a metric related to how much
a model warms globally into the base performance predictor set: annual-average SWCRE climatology in the Southern Hemi-
sphere midlatitudes, a region known for its reflective low clouds (Zelinka et al., 2020). Models that historically underestimate
Southern Hemisphere low cloud decks do not have them present to counteract future radiative warming increases associated
with the Hadley cell and its high cloud curtain moving poleward (Lipat et al., 2017; Tselioudis et al., 2016). Because European
change is superimposed on global change, models with these documented cloud cover biases should be penalized as well.

Model performance is benchmarked against predictors from the following observational datasets (Figure 5):

- SAT: Berkeley Earth Surface Temperature (BEST) merged temperature (Fig.5a,b; Rohde et al., 2013)

- SST: NOAA Extended Reconstructed Sea Surface Temperature version 5 (ERSSTv5, Fig.5c; Huang et al., 2017)

- PR: European-wide station data based E-OBS dataset (Fig.5d,e; Cornes et al., 2018)

- SWCRE: Clouds and the Earth's Radiant Energy System (CERES) Energy Balanced and Filled All- and Clear-Sky
  shortwave surface flux products (Fig.5f,g; Loeb et al., 2018, 2020)

- SLP: NOAA-CIRES-DOE 20th Century Reanalysis V3 reanalysis (Fig.5h; Bloomfield et al., 2018)

Here, we rely on a cosine-latitude-weighted RMSE distance metric computed between CMIP5/6 members and a single observa-
tional estimate for each predictor. Performance can also be based on more than one observational dataset per field (to represent
observational uncertainty), and as such, additional datasets can be included in future assessments to assess the sensitivity of
the selection protocol to the choice of observational dataset(s). The relationship between predictor RMSE from observed and
JJA CEU or DJF NEU SAT and PR change are shown in Supplementary Figures S6-S9. Distances for each predictor are subse-
quently normalized by dividing by their combined CMIP5/6 mean value, $\mu(\text{predictor}_{\text{CMIP5/6}})$, and averaged together to define
the "Aggregated Distance from Observed" performance metric for each model $i$ ($P_i$):

$$P_i = \frac{1}{6} \sum_{j=1}^{6} \frac{\text{predictor}_j}{\mu(\text{predictor}_{\text{CMIP5/6}})} \tag{1}$$

$$\tag{2}$$

Lower values of the metric, reflecting lower levels of model bias amongst the predictors, indicate higher performance. As a
reference, performance order in CMIP5 and CMIP6 for the two cases is presented in Supplementary Figure S10.





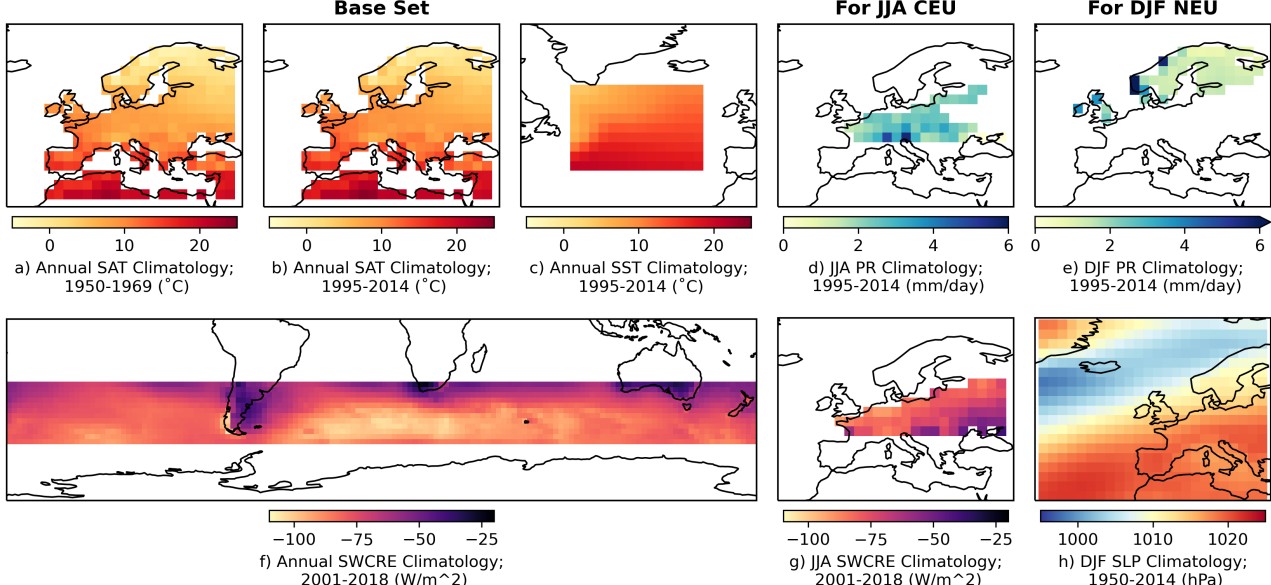

**Figure 5.** Observed predictor fields used to determine model performance for European climate applications in ClimSIPS; a base set used in both cases includes a) annual average Berkeley Earth Surface Temperature (BEST) European SAT climatology (1950-1969), b) annual average BEST European SAT climatology (1995-2014), c) annual average NOAA Extended Reconstructed Sea Surface Temperature version 5 (ERSSTv5) North Atlantic sea surface temperature (SST) climatology (1995-2014), and f) annual average Clouds and the Earth's Radiant Energy System (CERES) Southern Hemisphere midlatitude shortwave cloud radiative effect (SWCRE) climatology (2001-2018). For June-July-August (JJA) Central European (CEU) applications, d) JJA average E-OBS gridded Central Europe Station PR climatology (1995-2014) and g) JJA CERES CEU SWCRE climatology (2001-2018), are added to the base set. For December-January-February (DJF) Northern European (NEU) applications, e) DJF average E-OBS gridded Northern Europe Station PR climatology (1995-2014) and h) DJF NOAA-CIRES-DOE 20th Century Reanalysis V3 (NOAA-20C) North Atlantic sector sea level pressure (SLP) climatology (1950-2014), are added to the base set.

## 5.2 Spread in projected European temperature and precipitation change

Spread, the third and final dimension of ClimSIPS, differs from independence and performance because it is explicitly based on targeted future model outcomes rather than on historical model properties. While it is important for users to recognize that without independence, model agreement is meaningless, and without performance, uncertainty in future projections can be excessive, it is also important that users have the opportunity to sample novel climate outcomes if their application so requires. To allow users to maximize climate change signal diversity, we define spread as the distance between models in normalized JJA CEU and DJF NEU averaged SAT and PR change space, with change, as previously stated, referring to the difference between 2041-2060 and 1995-2014 mean state values in SSP5-8.5/RCP8.5. Normalization (subtracting the ensemble mean and dividing by the ensemble standard deviation) is carried out within CMIP5 and CMIP6 separately and ensures that SAT and PR distances





contribute equally to the spread metric $S_{ij}$. With normalized SAT and PR change for each model abbreviated as SAT$\Delta$ and
PR$\Delta$, respectively, spread distance between models $i$ and $j$ is:

$$S_{ij} = \sqrt{(\text{SAT}\Delta_i - \text{SAT}\Delta_j)^2 + (\text{PR}\Delta_i - \text{PR}\Delta_j)^2} \tag{3}$$

The only remaining complexity to computing spread is deciding on model representation in an ensemble where some models
contribute multiple members. Two strategies are explored. In the first, models with multiple ensemble members are represented
by their ensemble mean SAT and PR changes, alongside their individually represented counterparts. In the second, all models
are represented by an individual ensemble member chosen such that overall spread within the ensemble is at a maximum (i.e., is
farthest from all other members already placed in SAT-PR change space). We select spread-maximizing members from models
in a manner similar to the KKZ algorithm (Katsavounidis et al., 1994). First, all individually represented models are placed in
SAT-PR change space. Next, the model ensembles are assessed one by one and the member farthest from all already placed
models is chosen. Because member selection is done iteratively, there are multiple possible spread-maximizing solutions; here
we focus on one solution obtained by selecting from model ensembles in alphabetical order. Further details of individual
member selection are provided in Supplementary Section 5. We apply these two representation strategies to the performance
and independence metrics as well and enter into ClimSIPS with a set of 34 CMIP6 models (Table 1) and 26 CMIP5 models
(Table 2), each with a scalar performance score ($P_i$) and vectors of intermember (from Part I; $I_{ij}$) and spread ($S_{ij}$) distances
to all other models in the ensemble.

## 5.3   Cost Function and Subselection Triangle

With independence, performance, and spread metrics computed for each model, ClimSIPS can be carried out via a cost-function
minimization scheme. The first step of ClimSIPS is for the user to decide how many selections ($s_i$) they would like to make
from the total number of models in the selection pool ($N$). In this study, we demonstrate the method by selecting subsets
of varying sizes ($n$). To illustrate the method, we select two models from a purposefully reduced five model selection pool
($N = 5$, $n = 2$ or a "5 choose 2" subselection). We then explore method sensitivities and recommendation strategies with a 34
choose 3 subselection for CMIP6 Central European Summer case. Lastly, to suit a broader range of applications, we report
and recommend five model subsets for the Central European Summer and Northern European Winter cases from both CMIP5
($N = 26$) and CMIP6 ($N = 34$).

Once a subset size is decided upon by the user, ClimSIPS proceeds to compute the value of a cost function for each possible
combination of $n$ selections. Comprised of a performance term, $P(s_1, .. s_n)$, an independence term, $I(s_1, .. s_n)$, and a spread
term, $S(s_1, .. s_n)$, the cost function is:

$$C_{\alpha,\beta}(s_1, .. s_n) = (1 - \alpha - \beta) \cdot P(s_1, .. s_n) - \alpha \cdot I(s_1, .. s_n) - \beta \cdot S(s_1, .. s_n) \tag{4}$$

The importance to the user of $P(s_1, .. s_n)$, $I(s_1, .. s_n)$, and $S(s_1, .. s_n)$ are determined by two parameters, $\alpha$ and $\beta$. Both
parameters range from 0 to 1; $\alpha$ sets the importance of independence and $\beta$ sets the importance of spread. The importance of
performance, $1 - \alpha - \beta$ is a trade-off based on the importance of the other two terms that cannot be negative, thus requiring





that $\alpha + \beta \leq 1$. For each pair of $\alpha$ and $\beta$ values, there is a combination of models that minimizes the cost function based on their combined values of $P(s_1,..s_n)$, $I(s_1,..s_n)$, and $S(s_1,..s_n)$.

Because each model has a scalar performance score $P_i$, $P(s_1,..s_n)$ is defined as the sum of the normalized $P_i$ values in each subset:

$$P(s_1,..s_n) = \sum_{k=1}^{n} \frac{P_{s_k} - \mu(P_N)}{\sigma(P_N)} \tag{5}$$

$P_i$ values are normalized by subtracting the selection pool mean value $\mu(P_N)$ and dividing by the selection pool standard deviation $\sigma(P_N)$. The term is positive in the cost function because lower values of $P_i$ indicate smaller biases and thus higher performance. If a user prefers to select based on model performance only ($\alpha = 0$, $\beta = 0$), the set of $n$ highest performing models will minimize the cost function.

Model independence and spread metrics, $I(s_1,..s_n)$ and $S(s_1,..s_n)$, are based on the $I_{ij}$ and $S_{ij}$ distance matrices of the selected model subsets. The distances are normalized by the mean and standard deviation of their entire selection pool distance matrices ($I_{NN}$ and $S_{NN}$, respectively) and then summed over half of the matrix to avoid double counting:

$$I(s_1,..s_n) = \sum_{k<l}^{n} \frac{I_{s_k,s_l} - \mu(I_{NN})}{\sigma(I_{NN})} \tag{6}$$

$$S(s_1,..s_n) = \sum_{k<l}^{n} \frac{S_{s_k,s_l} - \mu(S_{NN})}{\sigma(S_{NN})} \tag{7}$$

In the cost function, $I(s_1,..s_n)$ and $S(s_1,..s_n)$ are negative terms because larger distances between models correspond to higher levels of independence and spread, which, along with higher performance, are the subset properties we prioritize. As independence and/or spread increases within a subset, the larger negative $I(s_1,..s_n)$ and $S(s_1,..s_n)$ terms eclipse the $P(s_1,..s_n)$ term, leading to a more and more negative minimum value of the cost function.

As previously discussed, different sets of models minimize the cost function for different values of $\alpha$ and $\beta$. To summarize how different subsets map to different priorities, we utilize a "subselection triangle" ternary contour plot (Harper et al., 2015). Ternary plots represent three-component systems that require the component contributions together to sum to a constant, typically 100%. The requirement, which reduces the degrees of freedom in the system from three to two, allows component value combinations to be plotted on an equilateral triangle with angled axes along each side. As the cost function balances the relative importance of independence ($\alpha$), performance ($1$-$\alpha$-$\beta$), and spread ($\beta$), it is an ideal candidate for such a visual representation.

We introduce the subselection triangle in a toy example of subselection: choosing two out of five CMIP6 models for JJA CEU applications (Figure 6). Of the five models in the selection pool, AWI-CM-1-1-MR is represented by its single member, while HadGEM3-GC31-MM, UKESM1-0-LL, MIROC-ES2L, and FGOALS-g3 are represented by their ensemble means. The models were chosen because they span a range of independence (Fig.6a), performance (Fig.6b), and spread (Fig.6c) within CMIP6 and therefore can help illustrate how the cost function selects subsets based on the different priorities. Intermember distances between the models in the selection pool (Fig.6a) highlight the low level of independence between family members




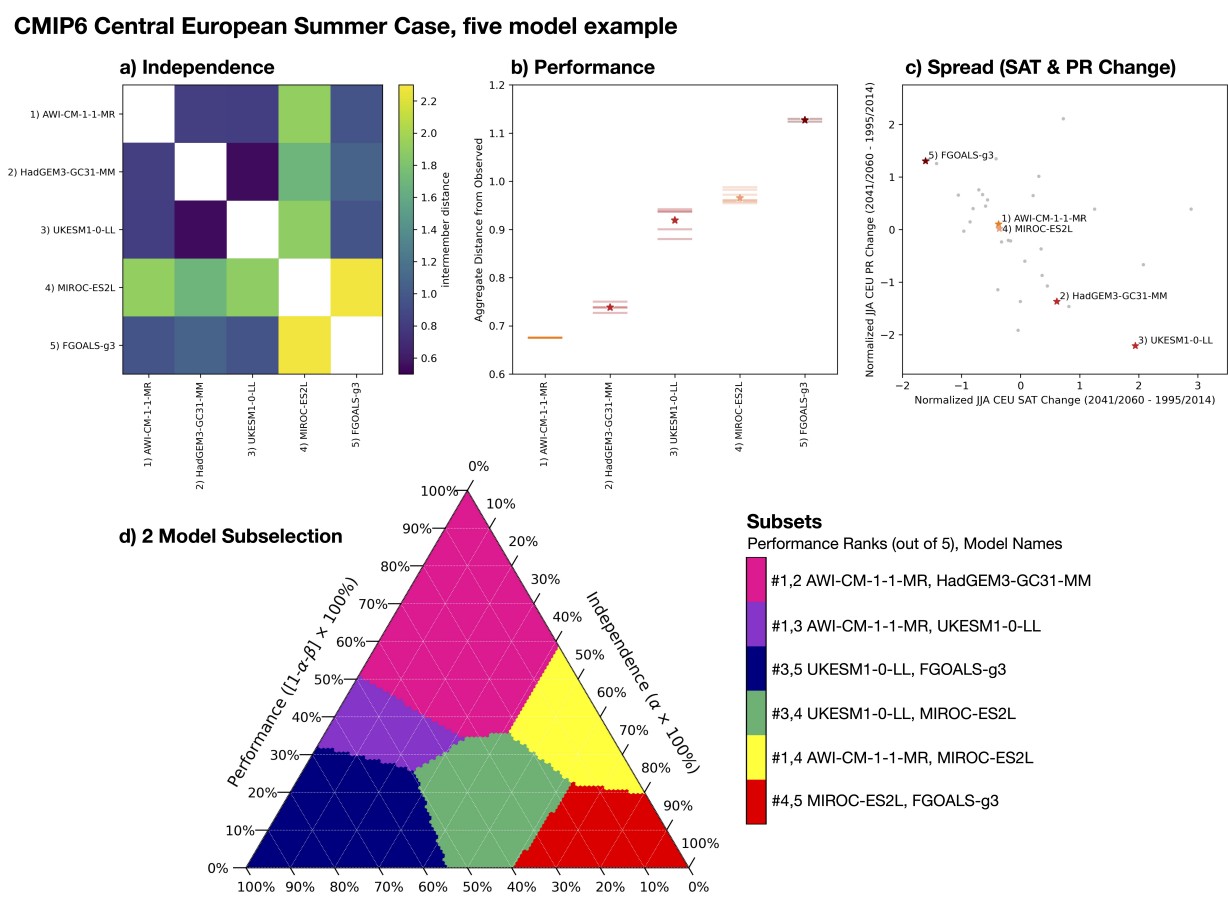

**Figure 6.** Five models from CMIP6 are used to illustrate ClimSIPS for JJA CEU applications. Independence, defined by intermember distances (Figure 3), for the five models are shown in panel a. Performance, defined as the average of model-observed RMSE for the six JJA CEU predictors (Figure 5), is shown in panel b. Lower values indicate a model is close to observed thus higher performing for the task. Performance of individual ensemble members are shown as horizontal lines and ensemble mean performance is starred. In panel c, the five selected models are highlighted among CMIP6 ensemble means (gray dots; single members where appropriate) in normalized JJA CEU SAT and PR change (SAT$\Delta$ and PR$\Delta$, respectively; 2041/2060 - 1995/2014 in SSP5-8.5) space. The target values are normalized by subtracting the CMIP6 mean and dividing by the CMIP6 standard deviation. Panel d shows the "subselection triangle" ternary contour plot. Selected subsets (colored regions) minimize a performance-independence-spread cost function as varying degrees of importance are placed on performance (1-$\alpha$-$\beta$), independence ($\alpha$), and spread ($\beta$). Subsets are listed by performance rank (out of five) and model name on the colorbar to the right of the triangle.

HadGEM3-GC31-MM and UKESM1-0-LL and the high level of independence of MIROC-ES2L with respect to all other models, as discussed in Part I. Performance scores, shown for each ensemble member (Fig.6b, horizontal lines) and ensemble means (Fig.6b, stars), lend themselves to an ordered rank of the models from #1 (AWI-CM-1-1-MR) to #5 (FGOALS-g3).



Placing each model in normalized SAT-PR change space (Fig.6c) demonstrates the spread in future climate outcomes possible within the selection pool (labelled stars) and with CMIP6 overall (gray dots). Notably, AWI-CM-1-1-MR and MIROC-ES2L, two relatively independent models, project near-identical changes in SAT and PR mean states by mid-century, while family members HadGEM3-GC31-MM and UKESM1-0-LL do not.

In Fig.6d, selected subsets are represented by colored regions of the subselection triangle. Boundaries between selected
subsets are determined by the distributions of performance, independence, and spread within the selection pool and by the step resolution on which $\alpha$ and $\beta$ vary. Throughout the study, we vary $\alpha$ and $\beta$ from 0 to 1 (in concert) in steps of 0.01. In the five model example, 6 two model combinations (out of a possible 10 combinations) minimize the cost function as $\alpha$ and $\beta$ vary in this way.

The subselection triangle is best investigated first along its boundaries. At each vertex, one property is given 100% priority,
and along each edge, only two priorities are balanced. At the top vertex, 100% priority is given to performance ($\alpha = 0$, $\beta = 0$), as indicated by the axis that runs down the left edge of the triangle. Along the left edge from top to bottom, the importance of performance diminishes while the importance of spread increases to reach 100% priority at the left vertex ($\alpha = 0$, $\beta = 1$). Along the bottom edge (the spread axis), the trade-off shifts to be between spread and independence, with independence being given 100% priority at the right vertex ($\alpha = 1$, $\beta = 0$). Finally, from bottom to top along the independence axis (the right edge
of the triangle), the importance of independence diminishes while the importance of performance increases.

Regions that intersect with vertices or edges can then be characterized according to priority. In Fig.6d, the magenta region that intersects with the triangle's top vertex is the subset of the highest performing models (Fig.6b), AWI-CM-1-1-MR and HadGEM3-GC31-MM, as indicated by the colorbar labeled with performance ranks out of five. From the point where performance and spread are each given 50% priority, AWI-CM-1-1-MR and UKESM1-0-LL minimize the cost function (purple
region). The blue region, intersecting with the left vertex, is comprised of the models furthest apart in normalized SAT-PR change space, UKESM1-0-LL and FGOALS-g3 (Fig.6c). In the green region, independence rather than performance takes priority alongside spread, resulting in a subset of the #3 and #4 ranked models, UKESM1-0-LL and MIROC-ES2L. Overall, if independence is prioritized, the cost function is likely to select subsets containing the independent MIROC-ES2L as seen in the red and yellow regions of the subselection triangle. For the yellow region, the balance between performance and independence
happens to yield a subset of models with little spread (AWI-CM-1-1-MR and MIROC-ES2L). Assigning no priority to a certain property does not necessarily mean the subset will be lacking in it; the other two subsets that fall along the independence axis (Fig.6 magenta and red regions) do have reasonable levels of model spread. However, without some level of priority given to a property, there is no guarantee it will be sufficiently represented.

## 5.4   Recommended Subsets for European Applications

Selecting more models from a larger selection pool leads to an increase in subsets composed of different models minimizing the cost function as $\alpha$ and $\beta$ vary. Subselection triangles become more complex and subdivided when three and five model subsets are selected from the full CMIP5 and CMIP6 ensembles; the six selected subsets from the 5 choose 2 CMIP6 JJA CEU example shown in Fig.6 become 37 selected subsets in the CMIP6 JJA CEU 34 choose 3 subselection by ensemble mean and 45



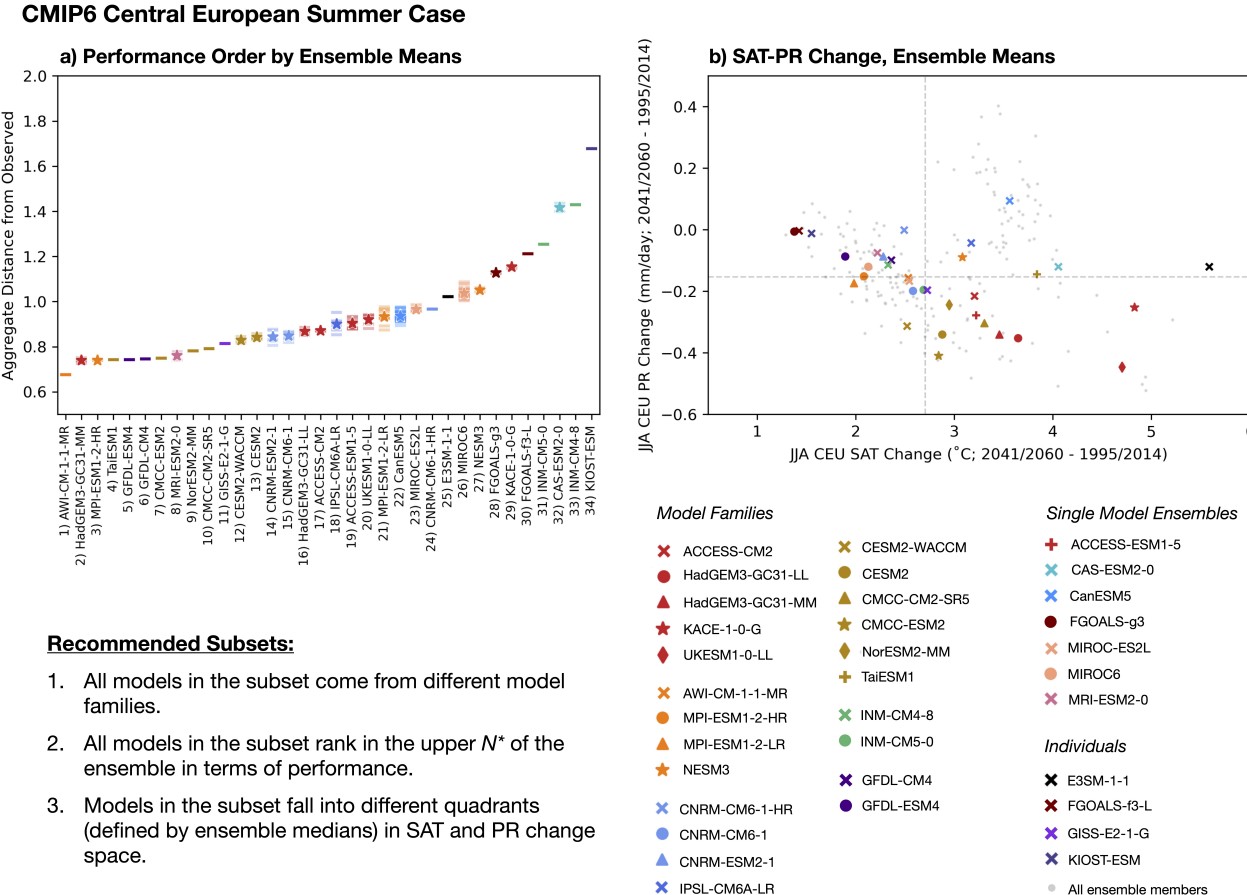

**Figure 7.** Independence, performance, and spread criteria that underpin model subset recommendations used throughout the remainder of the study. For independence, we recommend all models in a subset come from different model families (listed in the legend below panel b). For performance, we recommend all models in a subset rank in the upper $N^*$ of the ensemble with $N^*$ left up to user discretion. Performance in the full CMIP6 ensemble for JJA CEU applications is shown in panel a in terms of aggregated distance from observed, a metric based on the average of model-observed RMSE for the six JJA CEU predictors shown in Figure 5. Higher performers have a lower aggregated distance from observed and performance order is based on ensemble mean performance (stars) where applicable, i.e. when a model is represented by more than one ensemble member. Horizontal lines represent the performance of individual ensemble members. For spread, we recommend models within the subset fall into the different quadrants of JJA CEU SAT and PR change space. JJA CEU SAT and PR change (2041/2060 - 1995/2014 for SSP5-8.5) values for each ensemble member of CMIP6 (light gray dots) and for ensemble means (colored markers) are shown in panel b. Dashed gray lines indicate the median JJA CEU SAT and PR change value within the ensemble of ensemble means and separate the ensemble into four quadrants.

selected subsets in the 34 choose 5 subselection by individual member. To help users decide which subset is best suited to their
needs, we provide recommendations based on independence, performance, and spread criteria that go further than contribution



percentages. Recommendation criteria are listed in Figure 7 in conjunction with CMIP6 JJA CEU performance and (non-normalized) SAT-PR change distributions for models represented by ensemble means. The objective of these recommendation criteria is to further screen out selected sets that do not meet the following specific (user-defined) desirable properties. The criteria are:

1. All models in the subset come from different model families.

2. All models in the subset rank in the upper $N^*$ of the ensemble in terms of performance.

3. Models in the subset fall into the different quadrants (defined by ensemble medians) in SAT and PR change space. For subsets of four or more models, all quadrants must be represented.

For the independence criteria, model families defined in Part I are used as an independence guideline. For the performance criteria, the threshold $N^*$ is left up to user discretion; we choose different thresholds for the different regional/seasonal cases to accommodate the different performance distributions within CMIP5/6. For the spread criteria, we adapt the strategy of Ruane and McDermid (2017) and separate SAT-PR change space into quadrants with respect to ensemble medians. By recommending subsets of models that occupy the (relatively) cool/wet, cool/dry, warm/wet, and warm/dry margins of the ensemble, we ensure not only spread but a set of diverse future climate outcomes.

In Fig.7a, for models with multiple members, performance metrics are ordered by ensemble mean value (star markers), which are superimposed on the performance metrics of individual members (horizontal lines). The JJA CEU performance distribution in CMIP6 reflects that most of the models in the ensemble are not significantly biased with respect to observations, though several models are biased enough to form a perceptible tail. This implies that most CMIP6 models meet the basic standards to be considered suitable for downstream climate applications and allows us to set the $N^*$ threshold to be more inclusive than exclusive. Overall, individual member performance is tightly clustered around its ensemble mean for each model; this is true for all cases we explore in this study (see Sup. Fig. S10). The tight clustering confirms that performance is a model property rather than a member property. When defined by climatological predictors, performance is not just a matter of chance that observations match some members but not others due to internal variability. Tight clustering also means that representing models by ensemble mean versus an individual member will not fundamentally change performance order, aside from a few minor shifts up or down for models in the heart of the distribution.

In Fig. 7b, ensemble means (colored markers) and all CMIP6 ensemble members (gray dots) are placed within raw JJA CEU SAT-PR change space. Without normalization, it is clear that there is a wide range of JJA CEU SSP5-8.5 mid-century warming in CMIP6 models from 1.37°C to 5.59°C. However, models on both ends of the warming spectrum tend to be lower than average in terms of performance, suggesting these best- and worst-case warming projections may not be as realistic as the projections within the approximately 2-4°C warming range populated by the bulk of the CMIP6 ensemble. In terms of precipitation change, only CanESM5 (Fig. 7b, sky blue x marker) has an ensemble mean projecting wetter conditions in the region by mid-century; all other models project little change or an overall Central European summer drying. The joint SAT-PR change distribution is separated into a less warming/less drying quadrant containing ten models, a less warming/more drying



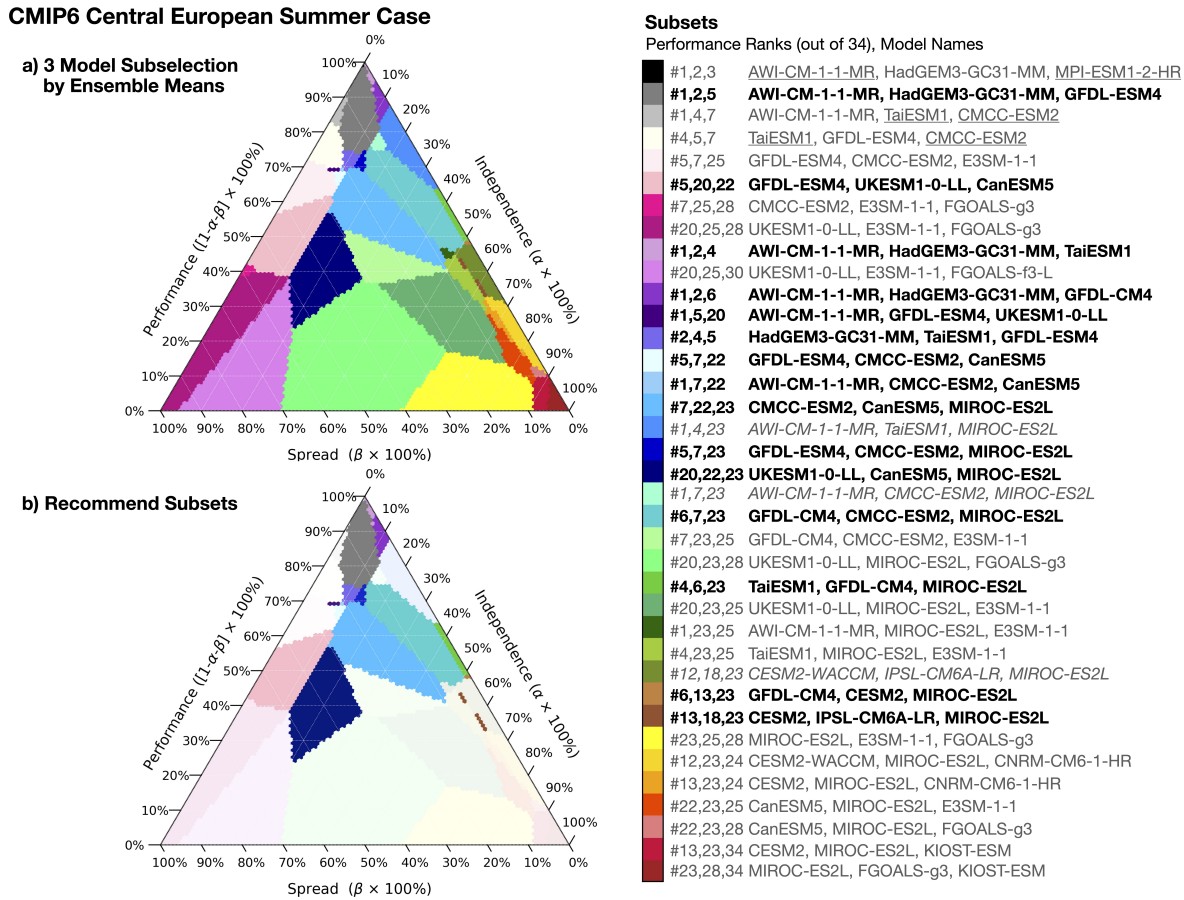

**Figure 8.** Ternary subselection triangles of the three model subselection from the full CMIP6 ensemble for JJA CEU applications. Each model is represented by its ensemble mean (or individual member when applicable). 37 three model subsets minimize the independence-performance-spread cost function for different value of $\alpha$ and $\beta$ (panel a). Models within each subset are listed with their performance rank (out of 34). Of the 37 possible subsets, we recommend 15 (panel b; labelled in bold) based on the following criteria: (1) all models in the subset come from different model families, (2) all models in the subset have a performance rank at or above 23 out of 34, and (3) all models in the subset come from different quadrants in JJA CEU SAT and PR change space. Remaining subsets that fail the performance criteria are listed in gray; those that fail the independence criteria are listed in gray with model family members underlined. Subsets that fail the spread criteria are indicated in italics.

quadrant containing seven models (including MPI-ESM1-2-HR which has a value slightly below the PR change ensemble
median), a more warming/more drying quadrant containing 11 models, and a more warming/less drying quadrant containing six models. Model labels, listed below Fig.7b, are grouped in terms of their designations determined in Part 1: as model families, single model ensembles, or individuals. In two of the six model families, all family members reside in the same quadrant; models with a MetUM-HadGEM3-GA7.1 atmosphere (Fig.7b dark red) all warm and dry more than the ensemble





median, while the two GFDL model variants both warm and dry less than the ensemble median (Fig.7b dark purple). The remaining model families each span two to three quadrants, demonstrating that model dependence is not necessarily clearly correlated with spread in projected outcome because internal variability/parameter perturbations influence the latter.

With independence, performance, and spread metrics defined, CMIP6 JJA CEU 34 choose 3 subselection, with models represented by ensemble mean, is presented in Figure 8. While stepping through $\alpha$ and $\beta$, 37 three-model combinations minimize the cost function; from the three highest performers (Fig.8a black region; AWI-CM-1-1-MR/HadGEM3-GC31-MM/MPI-ESM1-2-HR) to the three most independent models (Fig.8a dark red region; MIROC-ES2L/FGOALS-g3/KIOST-ESM) to the three models furthest from each other in SAT-PR change space (Fig.8a dark magenta region; UKESM1-0-LL/E3SM-1-1/FGOALS-g3). The subsets that minimize the cost function between those $100\%$ priority cases are labelled by model performance rank (out of 34) triplet and model names (Fig.8, colorbar). Note that model performance rank is used here specifically as a shorthand for comparing constituents across subsets, not as a commentary on how much better performing one model is than another model. Most models in the ensemble have a similar level of performance (Fig.7a), allowing us to set the recommendation performance threshold $N^*$ to include models in the top two-thirds of the ensemble, ranked up to and including #23 out of 34.

Applying all three recommendation criteria results in 15 recommended subsets (Fig.8b), which are listed as colorbar labels in black. For reference, all subsets recommended throughout the study are cataloged in Supplementary Tables S5 and S6. Of the remaining subsets, those listed in gray do not satisfy one or more of the recommendation criteria. Same model family representation within a subset is indicated with family members underlined. Subsets that include one or more model with a performance rank between #24 and #34 are listed in plain text gray, while subsets that have more than one model in the same SAT-PR change quadrant are listed in italics. There is a higher likelihood of selecting two models from the same model family in subsets comprised of high-performing models. Mirroring the finding of Sanderson et al. (2017) that high-performing CMIP5 models tended to be less independent and have more near replicates in the archive, the relationship between performance and independence we find in CMIP6 suggests that the global-scale SAT and SLP climatological differences that bestow independence also manifest as local and regional-scale model biases that diminish performance.

In the CMIP6 JJA CEU 34 choose 3 subselection by ensemble means, recommended subsets (Fig.8b) are comprised either solely of members of model families or a combination of model family members with the independent MIROC-ES2L and/or unique in SAT-PR change space CanESM5. Models with a CAM-based atmosphere (TaiESM1, CMCC-ESM2, and CESM2) are included in 10 of the 15 recommended subsets, GFDL variants (GFDL-CM4 and GFDL-ESM4) are included in nine, and MIROC-ES2L is included seven. Together, the three groupings form four of the 15 subsets. Models with a MetUM-HadGEM3-GA7.1 atmosphere (HadGEM3-GC31-MM and UKESM1-0-LL) and the ECHAM6.3-based AWI-CM-1-1-MR are also well-represented within the recommended subsets, appearing in six and five subsets, respectively. AWI-CM-1-1-MR and HadGEM3-GC31-MM tend to be chosen as family representatives because they are ranked first and second in performance in the ensemble. UKESM1-0-LL, which features the largest mid-century joint change in JJA CEU SAT and PR in the ensemble, takes over for HadGEM3-GC31-MM as its family's representative when the importance of performance drops below approximately $70\%$.

For use cases that require recommendations of specific simulations, we also run ClimSIPS with each model represented by an individual member. For subselection by ensemble mean, some models are represented by a member while others are





**Figure 9.** Three model subselection for CMIP6 JJA CEU applications with all models represented by an individual ensemble member. CMIP6 JJA CEU SAT-PR change (not normalized) for ensemble mean representation (as in Figure 7b) versus individual member representation are shown in panels a and b, respectively. Colored markers, indicating model representation, are superimposed on all CMIP6 ensemble members (light gray dots). Individual ensemble members are labelled in the key and were selected to maximize spread. In both panels, ensemble median values for JJA CEU SAT and PR change (gray dashed lines) delineate the four quadrants used for spread recommendations. The full and recommended (inset) subselection triangles for three model subselection by individual member are shown in panel c. As in Figure 8, recommended subsets are labelled in black. Subsets labelled in gray contain one or more models that (1) that come from the same model family (underlined), (2) fall below the performance threshold, or (3) that fall within the same quadrant of SAT-PR change space (italic). Additionally, common subsets in ensemble mean-based subselection (Fig.8) and individual member-based subselection share common colors; these subsets-in-common are also labelled with a starred performance rank triplet.



represented by an average. This representational difference is less of an issue for the independence and performance metrics,
which both are intentionally designed such that ensemble members do not deviate far from their ensemble mean values. For the
spread metric, however, an individual climate change projection and an average across a set of climate change projections are
unlikely to be equivalent due to internal variability. For example, a model's range of projected climate outcomes may include
interesting outlier cases that are curtailed by ensemble averaging. Using the ensemble mean as representation then does not
capture the model's full "spread potential" or ability to differentiate itself within the ensemble. For a model represented by only
one member, the projection provided as representation could fall anywhere within the model's un-sampled SAT-PR change
distribution; there is no way to know if it sits near to and thus reflects the model's hypothetical ensemble mean.

Figure 9 summarizes how individual member representation affects CMIP6 JJA CEU 34 choose 3 subselection, predom-
inately by amplifying model spread. When given the choice among ensemble members, the method selects outlier cases to
represent models to provide users interested in novel climate outcomes with not just a model, but a specific projection that may
be of interest. Compared to ensemble mean representation (Fig.9a), individual member representation (Fig.9b) provides an
increase, in CMIP6 JJA CEU PR change spread in particular, including an additional projection of wetter future JJA CEU con-
ditions (IPSL-CM6A-LR-r6i1p1f1) and fewer model overlaps in the core of the ensemble. There is not a substantial increase
in the range of JJA CEU SAT change in CMIP6, due to the fact that high and low change projections come from individually
represented models. To evaluate whether these differences were CMIP6 JJA CEU-specific, we also compare ensemble mean
representation to that of spread-maximizing members for the CMIP6 DJF NEU (Supplementary Figure S11), CMIP5 JJA CEU
(Supplementary Figure S12), and CMIP5 DJF NEU (Supplementary Figure S13) cases. Similar to the CMIP6 JJA CEU case,
CMIP5 JJA CEU PR change spread increases more than SAT change spread when models are represented by an individual
member versus an ensemble mean. In both CMIP6 and CMIP5 DJF NEU cases, however, the SAT change spread increase is
more striking than the PR change increase (Sup. Fig. S11).

The additional distance between models in SAT-PR change space afforded by individual member representation serves
to simplify the CMIP6 JJA CEU 34 choose 3 subselection triangle (Fig.9c). The number of cost function-minimizing model
subsets decreases from 37 subsets with 15 recommended when models are represented by ensemble mean (Fig.8a) to 21 subsets
with eight recommended when models are represented by individual member (Fig.9c). The two strategies share 12 subsets in
common, which are indicated by regions of common color in Figs. 8a and 9c and by starred model rank triplets in the colorbar
labels of Fig. 9c. Though some model ranks have shifted up or down due to the change in representation, the same set of 23
out of 34 models meet the performance threshold in both cases. Of the 12 shared subsets, two are recommended in both cases
(Figs. 8a,9c dark gray and teal), while three are recommended in ensemble mean subselection but not in individual member
subselection (Figs. 8a,9c dark sky blue, dark blue, and dark brown). This recommendation difference is due to MIROC-ES2L
shifting from less warming/more drying quadrant to the more warming/less drying quadrant when represented by member
r1i1p1f2 (Fig.9a,b salmon x marker). The remaining recommended individual member subsets feature many of the same
models/model families as in the ensemble mean case. A comparison of $I(s_1, ..s_n)$, $P(s_1, ..s_n)$, and $S(s_1, ..s_n)$ component
contributions to the minimized cost function, shown separately in Supplementary Figure S14, confirm the two representations
have qualitatively similar gradients in component magnitude within the subselection triangle. Because the two strategies yield





**Figure 10.** Ternary diagrams of the five model subselection for CMIP6 a) JJA CEU and b) DJF NEU applications. Each model is represented by an individual member, which are listed in the legend. Similarly to Figure 9, each case is shown alongside a subset recommendation triangle. Of the CMIP6 JJA CEU case's 45 possible subsets, six are recommended based on a performance rank threshold of 23 out of 34. For the CMIP6 DJF NEU case, five of 35 possible subsets are recommended based on a performance rank threshold of 31 out of 34.

similar results in our CMIP6 JJA CEU test case and because individual member representation has the additional advantage

of guiding users to specific simulations, we move forward to five model subselection by individual member for each of the European case studies, shown as subselection triangles in Figures 10 and 11.

Combinations of five models add further complexity to the CMIP6 JJA CEU subselection triangle (Figure 10a); 45 out of a possible 278,256 subsets minimize the cost function at different points in $\alpha$-$\beta$ space. We recommend six subsets based on the same recommendation criteria used in CMIP6 JJA CEU three model subselection with the additional condition that all four



**a) CMIP5 Central European Summer Case**

**b) CMIP5 Northern European Winter Case**

**Figure 11.** As in Figure 10, but for CMIP5 a) JJA CEU and b) DJF NEU applications. Of the CMIP5 JJA CEU case's 33 possible subsets, six are recommended based on a performance rank threshold of 17 out of 34 and a relaxed spread criteria (subset must span at least three quadrants). For the CMIP5 DJF NEU case, seven recommendations are made out of 45 possible subsets based on the same criteria as the CMIP5 JJA CEU case.

SAT-PR change quadrants are represented by the five models. All recommendations include AWI-CM-1-1-MR-r1i1p1f1 from the ECHAM6.3-based family (Fig.3a orange), one simulation with a MetUM-HadGEM3-GA7.1 atmosphere (HadGEM3-GC31-MM-r1i1p1f3 or UKESM1-0-LL-r1i1p1f2), and a representative from the CAM-based model family (Fig.3a gold; TaiESM1-r1i1p1f1 or CMCC-ESM2-r1i1p1f1). GFDL variants (Fig.3a purple) also appear in five of the six recommended subsets, suggesting that those four model families comprise a reasonable, independent subset spanning a range of climate out-



comes for CMIP6 JJA CEU applications. Depending on user needs, the highly independent MIROC-ES2L-r1i1p1f2, relatively less-biased MRI-ESM2-0-r1i1p1f1, or CanESM5-r16i1p1f1, which is one of few that project wetter future JJA CEU conditions should also be considered.

    Because region- and season-specific performance and spread metrics are used for each case, different CMIP6 subsets feature in the subselection triangle for Northern European winter applications (Fig.10b) than for Central European summer applica-

tions. Individual members chosen to represent models also differ between the DJF NEU and JJA CEU cases due to spread being case-specific. Of the 35 possible subsets that minimize the CMIP6 DJF NEU cost function, no subsets satisfy all three recommendation criteria as developed for JJA CEU applications. This is primarily due to models with performance ranks of 28 to 34 either being highly independent (e.g., #30 MIROC-ES2L-r9i1p1f2, #31 MIROC6-r12i1p1f1) or unique in pro- jected climate outcome (e.g., #28 E3SM-1-1-r1i1p1f1, #34 CAS-ESM2-0-r3i1p1f1, see Sup. Fig. S11b). In this instance, we

therefore chose to relax the performance threshold to consider models with a rank at or above 31 out of 34 to recommend five subsets. For CMIP6 DJF NEU applications, three simulations, CNRM-CM6-1-r5i1p1f2, CESM2-WACCM-r2i1p1f1, and MIROC-ES2L-r9i1p1f2, are included in all recommendations. When performance is given more priority (Fig.10b blue re- gions), the three are joined by AWI-CM-1-1-MR-r1i1p1f1 and a simulation with a MetUM-HadGEM3-GA7.1 atmosphere, either HadGEM3-GC31-MM-r2i1p1f3 or KACE-1-0-G-r3i1p1f1. This suggests again that a subset with the large model fami-

lies individually represented is a good starting point for downstream applications. When priority shifts towards independence and spread (Fig.10b green regions), subsets tend to include the models with the greatest CMIP6 DJF NEU mid-century positive and negative precipitation changes (E3SM-1-1-r1i1p1f1 and MIROC6-r12i1p1f1, respectively). Though not included in a rec- ommended subset primarily because of performance concerns, CAS-ESM2-0-r3i1p1f1 may also be of interest to some users in search of a CMIP6 DJF NEU worst case scenario; the simulation warms by 7.82°C between the 1995-2014 and 2041-2060

base periods (Sup. Fig. S11b).

    As a comparison to the CMIP6 cases, we also evaluate which five model subsets are selected from CMIP5 based on the same independence, performance, and spread definitions. An advantage of applying ClimSIPS to CMIP5 is that we can determine if it is able to select combinations of models recommended by EURO-CORDEX (CORDEX, 2018): NorESM1, MPI-ESM, HadGEM2-ES, GFDL-ESM, and EC-EARTH. Unfortunately, we were unable to include EC-EARTH in ClimSIPS due to

missing performance predictor fields but were able to consider the selection of NorESM1-ME-r1i1p1, NorESM1-M-r1i1p1, MPI-ESM-MR-r1i1p1, MPI-ESM-LR-r1i1p1, HadGEM2-ES-r4i1p1, GFDL-ESM2M-r1i1p1, and GFDL-ESM2G-r1i1p1 for subsets.

    In CMIP5 JJA CEU 26 choose 5 subselection (Figure 11a), 35 out of a possible 65,780 subsets minimize the cost function within the subselection triangle. Of the four EURO-CORDEX models, MPI-ESM variants (Fig.3b orange) do not appear in

any CMIP5 JJA CEU subsets selected by our cost function. The likely reason for this is that MPI-ESM-MR-r1i1p1 and MPI- ESM-LR-r1i1p1 are ranked #14 and #15 out of 26 in performance and tend to be relatively central in the ensemble in terms of both independence (Fig.3d) and spread (Sup. Fig. 12b). Therefore, when in combination with other models, the "mainstream" MPI-ESM variants do not have enough magnitude in their independence, performance, or spread metrics to help create a cost function minimum. The other three EURO-CORDEX models, however, do feature in selected subsets, notably in the





combination of NorESM1-ME-r1i1p1, GFDL-ESM2M-r1i1p1, and HadGEM2-ES-r4i1p1. The combination appears in two of
the five CMIP5 JJA CEU recommended subsets in the region of the subselection triangle where about 80% priority is given to
performance, 10% to independence, and 10% to spread.

Similar to the CMIP6 DJF NEU case, both CMIP5 JJA CEU and DJF NEU recommendations needed either a very lenient
performance threshold or a modified spread requirement for subsets to qualify. We chose to require models in CMIP5 subsets
to span at least three of the four SAT-PR change quadrants and to all have a performance rank at or above 17 out of 26. For
CMIP5 DJF NEU applications (Fig.11b), the relaxed spread requirement recommends seven of 45 possible subsets. While no
recommended subset includes a combination of the four EURO-CORDEX models, all include ACCESS1-0-r1i1p1, a close
relative of HadGEM2-ES, and CESM1-CAM5-r3i1p1, a successor of NorESM1. MPI-ESM and GFDL variants also appear
in several of the recommended subsets, suggesting that the independence, performance, and spread metrics we use are in line
with those used in other studies (e.g. McSweeney et al., 2015; Sanderson et al., 2017) and are therefore likely suitable for use
in CMIP6.

## 6     Discussion and Conclusion

In this study, we developed and demonstrated a method, ClimSIPS, to flexibly select subsets of CMIP models based on the
degree to which a user prioritizes model independence, model performance, and spread in projected climate outcomes. The
method is an extension of ClimWIP, a performance and independence weighting strategy pioneered in Knutti et al. (2017).
During the development of the ClimSIPS, we tested sensitivities and made several refinements to the definition of model depen-
dence in ClimWIP, which identifies model similarities via the absolute values of historical period, global-scale climatological
SAT and SLP predictor fields. Described in Part I, refinements included lengthening predictor climatological averaging periods
from 1980-2014 to 1905-2005 and designing predictor spatial fingerprints to explicitly reduce predictor spread within models
(e.g. amongst initial condition ensemble members) while preserving predictor spread between models. Computed separately
for SAT and SLP in CMIP5 and CMIP6, the fingerprints spatially masked the 15% of gridpoints where ensemble-wide internal
variability (defined as the median of the standard deviations within 12 CMIP6 and five CMIP5 initial condition ensembles) was
at a maximum, and the 15% of gridpoints where the traditional representation of between-model spread (defined as the standard
deviation across an ensemble comprised of one ensemble member per model) was at a minimum. An advantage of using cli-
matological SAT and SLP fingerprints rather than unmasked global fields to define model dependence was that masking helps
to future-proof against between-model convergence should model developers decide to tune, in particular, the absolute value
of global mean surface temperature in models (Mauritsen et al., 2012; Hourdin et al., 2017). Additionally, climatological SAT
and SLP fingerprints allayed a concern that computing RMSE distance between models does not require the overall collection
of intermember distances to meet the formal mathematical definition of metric space (Abramowitz et al., 2019). We found that
intermember distances within both CMIP ensembles did satisfy metric criteria, including the triangle inequality (dist(A,B) <=
dist(A,C) + dist(C,B)), and could therefore be both understood as distances and visualized in low-dimensional space.



Updates made to CMIP intermember distances assisted in our effort to make discrete delineations along the spectrum of dependence provided by ClimWIP. Three categories arose: single model ensembles, model families, and individuals. First and most dependent were single model ensembles comprised of multiple initial condition ensemble members, followed by those comprised of both perturbed physics and initial condition ensemble members. Next were model families, which we defined as self-contained groups in which all models were within a median intermember distance threshold and were closer to each other than to the rest of the ensemble. We were able to support all family designations with model descriptions and metadata; in CMIP6, model families emerged when models shared atmospheric components (e.g., MetUM-HadGEM3-GA7.1 or ECHAM6.3), developed from a shared atmospheric component (e.g., NCAR's CAM), or were variants from the same (e.g., GFDL, EC-EARTH, or INM) or closely collaborating (e.g., CNRM and IPSL) modeling centers. In CMIP5, similar model families were present, but with fewer models per family and fewer members per model than in CMIP6. Beyond model families, the last and most independent entities in CMIP were individuals or uniquely named models represented by a single simulation. The three categories formed a new "representative democracy" within CMIP, allowing us to explore how a stricter independence definition than the traditional "one model, one vote" requirement constrained distributions of ECS in CMIP5/6. By applying the new "one family, one vote" independence constraint, we saw CMIP6's bimodal ECS distribution shift and skew towards lower values of ECS, with the median and the 75th percentile each shifted by down by $0.43°C$ to $3.44°C$ and $4.29°C$, respectively. CMIP5 ECS, in contrast, maintained its raw distributional form under the one family, one vote independence constraint. Increased representation of certain model families from CMIP5 to CMIP6 explained part of the distributional difference in ECS between the two ensembles; restricting family over-representation reduced the median difference in CMIP5 and CMIP6 ECS by over $60\%$. We thus concluded that the increased ECS uncertainty range documented in CMIP6 is, in part, due to the fact that near-identical but differently named models appeared more frequently in CMIP6 and those models tended to have ECS values above $4.5°C$. Crucially, this conclusion could be drawn without any commentary on the quality of CMIP6 models, it simply rested on levels of model representation within the ensemble.

Leveraging the model dependence definition developed in Part I, we demonstrated ClimSIPS for summer and winter European case studies in Part II of this study. Performance was defined in terms of historical biases of concern rather than by historical strengths of unclear merit; we required models to effectively simulate annual climatologies of European SAT, North Atlantic SST, and Southern Hemisphere midlatitude SWCRE, in addition to local summer PR and SWCRE climatologies for Central European summer applications, and local winter PR and regional-scale winter SLP climatologies for Northern European winter applications. We found that most models are similarly (and not significantly) biased with respect to observations for the chosen climatological fields. However, in each case, a minority of models had aggregated distances from observed that were large enough relative to their peers to cast doubt on the projected future European climate states they simulated. These projected future climate states served as the basis of spread, which we defined using each model's JJA CEU or DJF NEU SAT and PR change between present (1995-2014) and midcentury (2041-2060) mean states. Because spread within CMIP5/6 depended on how a model with multiple ensemble members was represented, we explored subselection by ensemble mean and by an ensemble member selected to maximize CMIP5/6 spread overall. Depending on user needs, both spread representation





**Table 3.** Recommended CMIP6 five model subset by case and primary user priority

| **CMIP6 JJA CEU applications** | |
| --- | --- |
| Performance | AWI-CM-1-1-MR-r1i1p1f1, HadGEM3-GC31-MM-r1i1p1f3, MRI-ESM2-0-r1i1p1f1, TaiESM1-r1i1p1f1, GFDL-ESM4-r1i1p1f1 |
| Independence | AWI-CM-1-1-MR-r1i1p1f1, HadGEM3-GC31-MM-r1i1p1f3, MRI-ESM2-0-r1i1p1f1, TaiESM1-r1i1p1f1, MIROC-ES2L-r1i1p1f2 |
| Spread | AWI-CM-1-1-MR-r1i1p1f1, GFDL-ESM4-r1i1p1f1, CMCC-ESM2-r1i1p1f1, CanESM5-r16i1p1f1, UKESM1-0-LL-r1i1p1f2 |
| **CMIP6 DJF NEU applications** | |
| Performance | AWI-CM-1-1-MR-r1i1p1f1, HadGEM3-GC31-MM-r2i1p1f3, CNRM-CM6-1-r5i1p1f2, CESM2-WACCM-r2i1p1f1, MIROC-ES2L-r9i1p1f2 |
| Independence | CNRM-CM6-1-r5i1p1f2, CESM2-WACCM-r2i1p1f1, E3SM-1-1-r1i1p1f1, MIROC-ES2L-r9i1p1f2, MIROC6-r12i1p1f1 |
| Spread | AWI-CM-1-1-MR-r1i1p1f1, KACE-1-0-G-r3i1p1f1, CNRM-CM6-1-r5i1p1f2, CESM2-WACCM-r2i1p1f1, MIROC-ES2L-r9i1p1f2 |

strategies may be of interest, the former for studies requiring general model recommendations, and the latter for studies in need of specific simulations that project unique climate outcomes.

Subsets were then selected via a cost function, which optimized for sets of models that were more independent, higher performing, and more diverse in midcentury SAT and PR change. Computationally, the cost function was minimized with respect to all possible model combinations for each value $\alpha$ and $\beta$, parameters that determine the relative importance of independence, performance, and spread. Results were summarized in subselection triangles, ternary contour diagrams that showed which subset minimized the cost function for each value of $\alpha$ and $\beta$. On average, subselection of three to five models from CMIP6 yield around 20-45 possible subsets for a user to consider; as a guide, we offered recommendations of the model subsets that met additional qualitative independence, performance, and spread criteria. Recommended subsets were comprised of models that did not come from the same model family, all exceeded a performance threshold, and populated different quadrants of SAT-PR change space. Among the recommendations, the CMIP6 subsets of five individual spread-maximizing members that best prioritized independence, performance, or spread in our European case studies are listed below in Table 3.

We also subselected from CMIP5 to compare ClimSIPS subsets with the set of models recommended by EURO-CORDEX. We found that for the Central European summer case, our method selected three EURO-CORDEX recommended models (NorESM1-ME, GFDL-ESM2M, HadGEM2-ES) when $80\%$ priority was given to performance, $10\%$ to independence, and $10\%$ to spread. Of the remaining two models, EC-EARTH was not considered due to missing performance predictors, while MPI-ESM variants did not appear in any subset, likely due to performance, independence, and spread values that were relatively average within CMIP5 for the region and season. In the CMIP5 Northern European winter case, MPI-ESM variants did appear in subsets, but alongside CAM5-based CESM1-CAM5 rather than CAM4-based NorESM1 and ACCESS1-0 rather than its family member HadGEM2-ES. The appearance of EURO-CORDEX recommendations within the CMIP5 cases confirmed that our performance, independence, and spread definitions were in line with those used in other CMIP5 subselection studies and were therefore a suitable place to begin the conversation of CMIP6 subselection.

This study is meant to serve as a starting point for CMIP6 subselection, introducing a flexible subselection framework users can employ to decide for themselves which set of models best suits their specific needs. Subsets we recommended here are not the definitive answer for every climate application, but the method allows us to be transparent about our choices and to explore



the sensitivity of the result to those choices. By design, independence, performance, and spread metrics determine which models are selected by the cost function, and our defined metrics may be too general for some applications (e.g. hydrological modeling at the catchment scale). However, a strength of ClimSIPS is that it can incorporate any quantitative definition of independence, performance, and spread. The only requirements for use in the ClimSIPS cost function are that each model's
performance metric is represented by a scalar and that independence and spread metrics are defined between model pairs.

A potential limitation of ClimSIPS for some applications is high computational demand. Computing the cost function for all possible model combinations at each $\alpha$ and $\beta$ step (required to avoid incurring massive storage costs) may begin to become computationally burdensome when larger subsets are sought from larger selection pools. In our CMIP6 case studies, 34 choose 3 subselection computed the cost function 299,200,000 times (5,984 possible combinations $\times$ 50,000 $\alpha$ and $\beta$ steps), which
took approximately 68 minutes to run on a single core. 34 choose 5 subselection iterated over 16,231,600,000 cost function values, which, run in parallel, took approximately two hours to run on 24 cores. Not evaluated here, 34 choose 10 subselection, with the cost function computed $6.556407 \times 10^{12}$ times, would take considerably longer, an estimated three weeks to run, even in parallel on 48 cores. We contend that computational expense could limit some open use of the method, but users have several options that can alleviate the combinatorial explosion problem. First, the size of the selection pool can be reduced by
pre-filtering models that are highly dependent, low performing, or have convergent projected outcomes before computing the cost function. We intentionally did not pre-filter by independence, performance, or spread here though, because filtering is highly subjective and would be a disservice to users interested in subselecting from the whole ensemble. Second, users can compute the cost function for fewer values of $\alpha$ and $\beta$. This simplifying step will reduce the complexity of the subselection triangle, but potentially at the expense of some model combination minima.

In conclusion, ClimSIPS and its underpinning dependence definition provide users with a way to make systematic and intentional choices about the models they use. The method combines independence, performance, and spread considerations into a single optimization step for the first time and provides as output a transparent representation of the trade-offs between these three priorities. Such approaches are essential as CMIP archives grow and manual model selection becomes increasingly unfeasible.

*Code and data availability.* The code to generate figures in the main text and supplementary material is available as a collection of Python scripts at https://github.com/almerrifield/CMIP_subselection/releases/tag/v1.0 under the DOI https://doi.org/10.5281/zenodo.7492727 (Merrifield, 2022). Pre-processed input files for CMIP_subselection are available upon request. The ClimSIPS package is available at https://github.com/almerrifield/ClimSIPS/releases/tag/1.0.0) under the DOI https://doi.org/10.5281/zenodo.7668256 and is a part of the ETH Research Collection under the DOI https://doi.org/10.3929/ethz-b-000599363 (Merrifield and Könz, 2023). Both CMIP_subselection and Clim-
SIPS are made available under a GNU GPLv3 license. Pre-processed input files are provided for the ClimSIPS European case studies in the manuscript through the ETH Research Collection (https://doi.org/10.3929/ethz-b-000599312; Merrifield, 2023).



*Author contributions.* RK, RL, and LB conceived of the ClimWIP weighting scheme on which independence and performance metrics are based. ALM developed the model family and ClimSIPS code and analyzed the output, with contributions from VH. All authors contributed to the discussion of the methodology and results in the final paper. ALM wrote the paper with contributions from all co-authors.

*Competing interests.* The authors declare that they have no competing interests.

*Acknowledgements.* We would like to thank Erich Fischer, Benjamin Booth, and Karin van der Wiel for their helpful discussions during the development of the ClimSIPS. We would also like to thank Mario S. Könz for technical software engineering support, which helped bring the method into the realm of computational feasibility. This project was funded by the European Union's Horizon 2020 research and innovation program under grant agreements 820829 (CONSTRAIN) and 776613 (EUCP). We acknowledge the World Climate Research Programme's
Working Group on Coupled Modelling, which is responsible for CMIP, and we thank the climate modeling groups for producing and making available their model output.



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
