# Peer review of "Climate model Selection by Independence, Performance, and Spread (ClimSIPS) for regional applications"

_EGUsphere, 2022_

## Community Comment (CC1)

**Review of „Climate Model Selection by Independence, Performance and Spread (ClimSIPS) for regional applications" by Merrifield et al. (2023)**

by Swen Brands

General comment: The authors provide a comprehensive description of the ClimSIPS tool for weighting Global Climate Models according to the three criteria mentioned in the title. The study provides a detailed introduction to the concepts and methods used in this research field and comes with a detailed bibliography, so that it is almost a review article. The manuscript is well written, timely and relevant and I recommend a minor version in which the following points should be addressed:

1. Since the model independence results obtained in your study are very similar to those obtained in Brands (2022), the authors might wish to cite this study in the revised manuscript. Particularly the „one family one vote standard" (e.g. line 454) was also adopted in Brands (2022), where GCM clusters were built by combining the a priori criterion „use of the same AGCM family" based on Brands et al. (2023) with the a posteriori criterion „error pattern correlation coefficient > +0.65" (the Boé 2018 nomenclature is followed here). Albeit another predictor was used to measure a posteriori model dependence, the outcome is similar to yours (compare Figure 3 in Brands 2022 to your Figure 3). This shows that the results are robust to changes in the applied methodology.

2. Lines 53-56: „*modeling centers often contribute several versions of their base model under different names as well (Leduc et al., 2016); these variants differ by, for example, the spatial resolution of some model components or biogeochemical cycling, which may influence their simulated climate in ways that are difficult to anticipate.*"

Should start with an uppercase letter ("Modeling centers..."). Model versions from the same center often differ by the inclusion of entire numerical sub-models describing specific Earth system components in addition to the basic four components atmosphere, land-surface, ocean and sea-ice. In this context it is interesting to note that the names and versions of all sub-models representing up to 12 climate system components is listed in Brands et al. (2023) for 61 nominally distinct GCMs from CMIP5 and 6. In this extensive metadata archive, you can see how the distinct modeling groups have built their models in terms of included sub-models / Earth System components.

2. Lines 56-57: „*Adding further complexity, even uniquely named models from different modeling centers fall along a spectrum of uniqueness.*"

I do not fully understand what you mean with this sentence. Could you provide an example for „*uniquely named models from different modeling centers*" ?

3. Lines 58-61: I think the Brands et al. (2023) GCM metadata archive is relevant in this sentence as well and the authors might wish to refer to it. The archive could be alternatively cited in lines 78-80 and is useful for determining „a priori" dependencies within in the CMIP ensemble, as defined by Boé (2018).

4. Lines 119-121. Meanwhile, the EURO-CORDEX model selection team has come to a final recommendation for the driving GCMs from CMIP6. Please see Sobolowski et al. (2023) for more details.

6. Lines 205-217: Please indicate the time aggregation of the GCM and reanalysis data you are using. Is the study based on monthly-mean data?

7. Lines 218-219 and elsewhere: Would make sense to use the terms „a priori" and „a posteriori" model dependence (Boé 2017) in this study?

8. Lines 230-238: The definition of the INV and SME groups is clear but more information is needed on how you define the FAM group. For example, ACCESS-ESM1-5 is here considered an „SME" model, meaning that it „*[...] is represented by multiple members (e.g., initial condition ensembles, perturbed physics ensembles, combinations thereof) but is not determined to be part of a broader multi-model family.*"

However, a closer look at the „source" attributes of the corresponding netCDF files from ESGF and at the reference articles (doi: 10.1071/ES19035, 10.5194/gmd-12-4999-2019, 0.5194/gmd-4-723-2011,0.5194/gmd-4-1051-2011) reveals that the entire ACCESS GCM family is based on versions of the atmospheric sub-models (or AGCMs) developed at the MetOffice-Hadley Centre. Namely, ACCESS-ESM1-5 makes use of the „HadGAM2" AGCM that is also used by the HadGEM2-ES and HadGEM2-CC coupled model configurations. HadGAM2 was further developed into „MetUM-HadGEM3-GA7.1", constituting the AGCM used in both Hadley Centre's and CSIRO's coupled model configurations used in CMIP6, e.g. HadGEM3-GC31-MM (doi: 10.1071/ES19040) and ACCESS-CM2 (doi: 10.1071/ES19040). Thus, it is reasonable to put the HadGEM and ACCESS coupled model configurations to the same family, as was done in Brands (2022), because they essentially share their atmospheric component. Following your nomenclature, this would mean assigning a „FAM" to ACCESS-ESM1-5. Note that all the aforementioned model metadata is available at one glance from Brands et al. (2023).

9. Lines 240-247: Could you also shortly refer to the disadvantages of the a posteriori / output data – driven approach to measure GCM dependence ? Here, only the advantages are described so far.

10. Lines 259-261: Please add an equation to define inter-member GCM distance.

11. Lines 279-280: The observational density underlying theses gridded dataset is also reduced during the first half of the 20th century, particularly in the Southern Hemisphere.

12. Lines 285-287: „relative change with respect to a historical period" is not considered „model performance", as far as I know. Traditionally, the term „model performance" refers to model error with respect to observations.

13. Line 307: „confined to subtropical regions" > „confined to the tropics and subtropics"

14. Lines 323-324: The Brands et al. (2023) metadata archive comprising names and versions of the sub-models used in each GCM configuration helps to identify the „very similar but differently named models" you refer to in this sentence.

15. Figure 3b) I can here see 3 independent clusters instead of the 2 indicated in the caption.

16. Between page 16 and 17 it seems that some running text is missing.

17. Lines 434-436: Similarities might be caused by the use of similar ocean models. Distinct versions of the same OGCM (NEMO) are used in the CNRM and IPSL GCMs (see Brands et al. 2023 for further details).

**References**

Boé, J. (2018). Interdependency in multimodel climate projections: Component replication and result similarity, *Geophysical Research Letters*, 45, 2771– 2779. https://doi.org/10.1002/2017GL076829

Brands, S. (2022). Common error patterns in the regional atmospheric circulation simulated by the CMIP multi-model ensemble. *Geophysical Research Letters*, 49, e2022GL101446. https://doi.org/10.1029/2022GL101446

Brands, S., Tatebe H., Danek, C., Fernández, J., Swart, N. C., Volodin, E., Kim, Y.H., Collier, M., Bi, D., Tongwen, W. (2023). SwenBrands/gcm-metadata-for-cmip: First standalone version of GCM metadata archive "get_historical_metadata.py" (v1.1). Zenodo. https://doi.org/10.5281/zenodo.7813495

Sobolowski, S. et al. (2023). EURO-CORDEX CMIP6 GCM Selection & Ensemble Design: Best Practices and Recommendations. Zenodo. https://doi.org/10.5281/zenodo.7673400

---

## Author Comment (AC1)

The authors describe an approach for analysis of inter-dependence of CMIP climate model simulations, their weighting and sub-selection for different purposes based on desired range of spread, performance and dependency.

The manuscript is well written and fits the scope of GMD. I find the developed methodology innovative and useful.

One of the important findings is that the model performance is rather a "model" characteristic, whereas the spread is more diverse for individual members of an ensemble of the same model. Further, the reduction of the spread of ECS after the family-democracy is taken into account is also a very important conclusion.

Please find below comments that should be addressed before the paper is accepted for publication:

We'd like to thank you for taking the time to help us improve our manuscript; we are thrilled to hear you find the methodology useful! All comments have been addressed in the text and as indicated below for quick reference.

line 234 – 237: I suggest explaining better that the "multi-model ensembles" correspond to "families", e.g. replacing the word "ensembles" with "families".

Thank you for the catch, we've changed ensembles to families to maintain consistency.

L209-212 : "In total, the 218 CMIP6 simulations from 37 uniquely named models considered in Part I fall into 19 Groups (7 multi-model families, 8 single model ensembles, and 4 individuals) and the 75 CMIP5 simulations from 29 uniquely named models fall into 20 Groups (8 multi-model families, 5 single model ensembles, and 7 individuals)."

line 260-265: the results of the sensitivity testing are shown somewhere? it should be stated explicitly (e.g. "see below")

This is a good point. We've decided to make the sensitivity testing more transparent by including an additional figure in the Supplementary Material. In the main text, we've updated the following in reference to the new figure:

L246-248 : "To first order, $I_{ij}$ is robust to methodological choices; the sensitivity testing did not reveal major shifts in whether a model was considered relatively dependent or independent with respect to the other models in the ensemble (See Figure 1, Supplementary Figure S1)."

L308–310 : "Results are not highly sensitive to precise percentile thresholds used to exclude regions of low between-model spread and high within-model spread; intermember distances are largely consistent for thresholds between the 5th and 20th percentile for between-model spread and the 80th and 95th percentile for within-model spread (Sup.Fig. S1)."

To the Supplementary Material, we've added:

L15-23: "While developing the fingerprint mask, we explored sensitivities to the percentile thresholds that define "low" between-model spread and "high" within-model spread. Shown in Supplementary Figure S1, we varied the threshold to mask between-model spread at or below the 5th, 10th, 15th, and 20th percentile. In concert, within-model spread was masked at or above the 95th, 90th, 85th, and 80th

*percentiles. Intermember distances were similar in the four cases. They primarily differed by how closely members of initial condition ensembles group together. Ultimately, we chose the 15th and 85th percentile thresholds to define independence but would have obtained similar results with the 20th and 80th percentile thresholds. However, we felt that masking 40$\%$ of the domain began to challenge the notion of global similarity in the independence predictor fields and thus moved forward with the 15th and 85th percentile thresholds."*

*The new figure is captioned as follows:*

[Figure]

*Supplementary Figure S1: "A comparison of CMIP6 intermember distance sensitivity to the definition of "low" between-model spread and "high" within-model spread. Regions at or below/above the following percentile thresholds are masked: below the 5th and above the 95th (panel a), below the 10th and above the 90th (panel b), below the 15th and above the 85th (panel c, used in the study), and below the 20th and above the 80th (panel d). For each model, distances between initial condition or perturbed physics ensemble members are marked in color, and distances to members of the remaining models are marked in light gray."*

line 270 – 274: I suggest shortly mentioning that the benefit of longer time period is not visible for all models, denoting the contradictory result of EC-EARTH3 models, for which there is still the overlap even for the longer period.

We agree that this is the right place to identify what is happening with EC-Earth3. We've amended the text to read:

*L254-263: "The grouping effect of the longer predictor averaging period helps to further distinguish initial condition / perturbed physics ensemble members from members of other models (Fig.1, light gray) in most cases. This differentiation is particularly clear in the case of CESM2-WACCM. The longer climatological averaging period distinguishes its three ensemble members from those of CESM2; with the shorter period, the two CESM2 model variants overlap (Fig.1, models 11 and 12). In contrast, though, the longer averaging period fails to subdue internal variability enough to differentiate EC-Earth3-Veg from its base model, Earth3 (Fig.1, models 23 and 24). The remaining internal variability in EC-Earth's global SAT and SLP fields is traceable to oscillations in the EC-Earth3 preindustrial control run from which both model variants are branched (Döscher et al. 2022). Functionally, this means that despite differing by coupled dynamic global vegetation, EC-Earth3 and EC-Earth3-Veg would be identified as one model by our independence metric."*

line 294: I suggest adding a note that the concept of fingerprints will be explained further below.

This is a good point. To clarify the terminology "fingerprint", we've reworded the passage to:

*L282-283: "Further, spatial masks can be explicitly designed to leave behind "fingerprints" tailored to meet dependence objectives. Here we design a spatial fingerprint..."*

line 546 – 547: why the evolution of SAT over Europe should be representative of the GCM's ability to simulate correctly the response to aerosol forcing? There are also other factors to be taken into account, so why specifically only the aerosol emissions are mentioned here?

Thank you for bringing this to our attention, the way that we've worded things really overstates our rationale for including two SAT climatological periods. The idea more was to evaluate to what extent a model resembled observed European SAT during a period prior to and post- the adoption of air quality directives in Europe to ensure biases during either period were accounted for in the performance metric. We did not explicitly evaluate how models respond to aerosols and should not imply that. Therefore, we've changed the sentence to read:

*L536–538 : "We employ two periods of annual-average European SAT climatology, 1950-1969 and 1995-2014, to establish (1) if notable European SAT biases exist in the period prior European air quality directives (Haug et al. 2004) and (2) if a model's "present day" European SAT is significantly warmer or cooler than observed."*

line 608: I recommend to explain a bit the term "pool" – it can be the whole multi-model ensemble or somehow pre-selected subset. The term pops-up suddenly and makes the reader a bit confused.

This is a good point. We've decided to adapt the notation over all as follows:

*L601–603 : "The first step of ClimSIPS is for the user to decide the number n of selections ($s_i$) they would like to make from a selection pool of N available models ($s_1,..s_N$). In this study, we demonstrate the method by selecting subsets of varying sizes from selection pools of varying sizes, henceforth referred to as a "N choose n subselection"."*

In general, we feel that the $s_i,...,s_N$ notation helps to highlight instances where we normalize based on the whole selection pool. This improves interpretability in equations 7-9.

lines 612-620: the notation "$s_i$" should be explained properly, that it denotes individual simulations.

We also hope the adaptation of the notation from the selection pool being represented by N to it being represented by $s_i,..s_N$ helps here some. In addition, we've added the following to indicate that $s_i$ refers to individual simulations:

L603–605 : "To illustrate the method, we select two model simulations, $s_1$ and $s_2$ from a purposefully reduced five model selection pool, $s_1,..s_5$, in a 5 choose 2 subselection."

line 875: please add a reference to the proof of the statement "intermember distances within both CMIP ensembles did satisfy metric criteria". (is it shown somewhere or not shown?)

This statement is rather abrupt and definitely comes too late. To address this, we've added the following to the method description:

L332-338 : "In Figure 3, we show how intermember distances based on the sum of normalized RMSEs calculated from SAT and SLP fingerprints help to uncover model relationships within CMIP. Intermember distances are presented for each model in one dimension (Fig.3a,c) and, as recommended by Abramowitz et al. (2019), for the ensemble as a whole in a low dimensional projected space (Fig.3b,d). The second display strategy is appropriate because we find our intermember distance matrix meets the formal mathematical definition of a metric space. To be mathematically a metric, the distance from a model to itself must be zero, and distances between models must be positive, symmetric, and adherent to the triangle inequality, which states that the distance from A to B is less than or equal to the distance through an intermediary point C (Abramowitz et al. 2019)."

We then call back to the explanation with:

L856–860: "Additionally, climatological SAT and SLP fingerprints allayed a concern that computing RMSE distance between models does not require the overall collection of intermember distances to meet the formal mathematical definition of metric space (Abramowitz et al. 2019). We found that intermember distances within both CMIP ensembles did satisfy metric criteria, with all sets of three models upholding the triangle inequality of dist(A,B) <= dist(A,C) + dist(C,B). Intermember distances could therefore be both understood as distances and visualized in low-dimensional space."

Part II – a comment on ternary plots and recommended subsets: The ternary plots are definitely useful for the analysis of different selection criteria. An issue, that is not commented on, is that some of the subsets "reside" a large part of the triangle, whereas some other subset have only a small fraction of the triangle. In some cases, the subset minimizes the cost function for only very narrow intervals of the coefficient values.

Complexity is an interesting feature of the subselection triangles, and we agree that it is worth discussing further. The size of the selection region is determined by the distributions of the selection criteria. In our primary JJA CEU case study, all three selection criteria have more or less a "core" with a few outliers (e.g., MIROC6 and MIROC-ES2L for independence or E3SM-1-1 and CanESM5 for spread). Thus, to first order, "small" regions of the selection triangle tend to occur when balancing performance and independence (with 10% or less priority given to spread) or when performance priority begins to give way to the other two (~70% performance priority). In the first instance, many models have similar performance

values and can be selected alongside the independent MIROC models to minimize the cost function as performance priority gives way to independence priority. In the second instance, performance priority no longer requires the highest performing model to be included and subsets are comprised of other constellations of models until spread maximizing and more independent models eclipse them in the cost function. "Large" regions of the triangle tend to occur once performance is not a key player anymore and nearly always involve the independent MIROC models or the spread-maximizing models like E3SM-1-1 or CanESM5. Because those models stand away from the core to such a degree, they minimize the cost function for large regions of the triangle. In short, the distribution of selection criteria matters and outliers create larger regions. Because of CMIP6's "core", the triangle is more complex.

As a discussion, we've added the following. To discuss region size in general:

*L687–691: "The size, shape, and number of regions within the subselection triangle are determined by performance, independence, and spread distributions; the larger the selection pool, the more difficult it becomes to predict the combination of models that will minimize the cost function. A subset can minimize the cost function for a small region in α-β space or even a single value of α and β. Small subset regions are as valid as larger ones; they simply reflect that independence, performance, and spread are distributed such that there are several model combinations in contention to minimize the cost function in that region of the subselection triangle. Conversely, when a subset minimizes the cost function for a large region of the subselection triangle, it suggests that it is comprised of outliers given priority in the cost function to such an extent that other model combinations cannot reach the minimum."*

To highlight region size relative to the whole domain:

*L756: "In total, recommended subsets cover 15% percent of the subselection triangle."*

To discuss small recommended regions specifically:

*L764-769: "Small recommended subset regions (<10 pixels in α-β space) occur at approximately 70% performance, 10% independence, and 20% spread, likely because performance priority has reduced enough to allow spread outliers like UKESM1-0-LL and CanESM5 to be in contention alongside various models within the core of the performance distribution. Similarly, small recommended subset regions near 50% performance and 50% independence result from the selection of various models in the performance core with the independent MIROC-ES2L."*

I suggest that it should be discussed, that in the case of the subsets that correspond to a very small fraction, there might be other subsets that have cost function values close to minimum and would maybe satisfy the criteria for a wider interval of the coefficients?

We looked into this by looking at the cost-function's secondary minimum, i.e., the subset of models that is next in line to minimize the cost-function. For the JJA CEU 35 choose 3 by ensemble means case shown in Figure 8, the primary (top) and secondary (bottom) minimum subselection triangles are shown below:

[Figure]

The several pixel small regions that were recommended are highlighted. In all cases, they are expanded (in pixel / % of domain) by the secondary minimum as:

- AWI-CM-1-1-MR, CMCC-ESM2, and CanESM5
  - (1 / 0.01%) > (8 / 0.08%)
  - This subset has a similar cost function to:
    - CMCC-ESM2, GFDL-ESM4, and CanESM5
- AWI-CM-1-1-MR, GFDL-ESM4, and UKESM1-0-LL
  - (3 / 0.03%) > (40 / 0.39%)
  - This subset has a similar cost function value to:
    - CMCC-ESM2, GFDL-ESM4, and TaiESM1
    - CMCC-ESM2, E3SM-1-1, and GFDL-ESM4
- CESM2, GFDL-CM4, and MIROC-ES2L
  - (2 / 0.02%) > (5 / 0.05%)
  - This subset has a similar cost function to:
    - CMCC-ESM2, GFDL-ESM4, and MIRCOC-ES2L
    - CESM2-WACCM, IPSL-CM6A-LR, and MIRCOC-ES2L
- CESM2, IPSL-CM6A-LR, and MIRCOC-ES2L
  - (8 / 0.08%) > (81 / 0.79%)
  - This subset has a similar cost function value to
    - CESM2-WACCM, IPSL-CM6A-LR, and MIRCOC-ES2L

Several larger regions are also expanded by the secondary minimum, but it is not that the cost function minimum is unstable and the secondary minimum is stable, unfortunately.

Could there be some additional selection criteria that the recommended subsets should minimize the cost function for a larger fraction of the ternary plot?

This is an interesting idea and we are considering adding some penalties to, for example, remove subsets that include more than one family member from contention. An option to pre-filter by performance is already implemented. We are not sure how these sorts of penalties will affect complexity of the subselection, but with ClimSIPS, it is straightforward to explore!

It would make the selection more robust. In some cases, the recommended subsets are represented by only several "points" in the ternary plot (e.g. Fig 8). I believe that it is desirable to recommend subsets that would be useful for as wide range of applications as possible, to make projections used for similar applications physically consistent.

As far as robustness of the selection, we do feel that the beauty of CMIP is there is not a one size fits all answer for every study. There are just a few "good" models to use in all cases. As priorities shift, so do the combinations of models. But we do agree, physical consistency for similar applications is important as well. Alternative definitions of performance or spread in other variables could help coalesce subset regions.

Part II + Discussion and conclusion: Regarding the recommended subsets derived from CMIP5, the ClimSIPS method suggests similar subsets as used in Euro-CORDEX for driving regional climate model simulations over Europe. The authors claim, that this agreement implies, that their method is suitable for choosing subsets for driving RCMs. This implication is questionable, as it is not clear, what exactly was the basis for the choice of Euro-CORDEX driving GCMs from CMIP5. I do not doubt that proposed method is suitable for choosing appropriate subsets from CMIP6, I just do not agree with the comparison to CMIP5 subsets implying the suitability of ClimSIPS. Please, consider modifying the statements appropriately. The argumentation should be based on the nature of ClimSIPS, which is well described in the paper.

Thank you for bringing this up, and we are very happy to hear you find the method suitable on its face. Following this feedback, we've decided to move away from the CMIP5 subselection and only focus on CMIP6 subselection in the main text. CMIP5 subselection will remain for those interested in the supplement. We feel this move has made a very long paper a bit more manageable to read. The statements you mention have been removed from the main text and text descriptions in the supplement. Thank you again for your careful read of the study!

Language, copy-edits:

line 85 – Sentence beginning „Initial versions ..." – the verb is missing in the sentence.

Thank you, fixed.

L72–73 : "Initial versions of ClimWIP based performance and independence definitions on the same set of predictors, which lead to concerns about convergence to reality."

line 149 – „The study, an extension **of** the work ..." – the "of" is missing

Thank you, fixed.

line 508 – "in" is missing in "For use **in** cases..."

Thank you, fixed.

---

## Author Comment (AC2)

**Review of „Climate Model Selection by Independence, Performance and Spread (ClimSIPS)
for regional applications" by Merrifield et al. (2023) by Swen Brands**

General comment: The authors provide a comprehensive description of the ClimSIPS tool for
weighting Global Climate Models according to the three criteria mentioned in the title. The study provides a
detailed introduction to the concepts and methods used in this research field and comes with a detailed
bibliography, so that it is almost a review article. The manuscript is well written, timely and relevant and I
recommend a minor version in which the following points should be addressed:

Dr. Brands, thank you so much for taking the time to review our manuscript. That
you find it almost a review article is a high compliment, especially in comparison
to your well-cited studies. We've worked to carefully address your review point-
by-point and feel that it has improved the manuscript. We hope you feel the same.

1.  Since the model independence results obtained in your study are very similar to those obtained in
    Brands (2022), the authors might wish to cite this study in the revised manuscript. Particularly the
    „one family one vote standard" (e.g. line 454) was also adopted in Brands (2022), where GCM clusters
    were built by combining the a priori criterion „use of the same AGCM family" based on Brands et al.
    (2023) with the a posteriori criterion „error pattern correlation coefficient > +0.65" (the Boé 2018
    nomenclature is followed here). Albeit another predictor was used to measure a posteriori model
    dependence, the outcome is similar to yours (compare Figure 3 in Brands 2022 to your Figure 3). This
    shows that the results are robust to changes in the applied methodology.

    It is fantastic to see that our approach is more or less consistent with
    field standards! (We've been working to develop the ClimWIP independence
    definition to be consistent with known dependencies for several years.)
    We've added a few references to Brands 2022 throughout the paper including:

    L382–384: "We also anticipated three "extended" families based on an
    analysis of model metadata, summarized in Sup. Tabs. S1 and S2, and the
    work of Brands (2022b), which grouped models in CMIP5 and CMIP6 via shared
    atmospheric circulation error patterns."

    L866–867: "We were able to support all family designations with model
    descriptions and metadata and found our designations to be broadly
    consistent with other model output and metadata-based dependence
    definitions (Brands, 2022)."

2.  Lines 53-56: „modeling centers often contribute several versions of their base model under different
    names as well (Leduc et al., 2016); these variants differ by, for example, the spatial resolution of some
    model components or biogeochemical cycling, which may influence their simulated climate in ways
    that are difficult to anticipate."

    Should start with an uppercase letter ("Modeling centers...").

    Thank you; fixed.

    Model versions from the same center often differ by the inclusion of entire numerical sub-models
    describing specific Earth system components in addition to the basic four components atmosphere,
    land-surface, ocean and sea-ice. In this context it is interesting to note that the names and versions of
    all sub-models representing up to 12 climate system components is listed in Brands et al. (2023) for
    61 nominally distinct GCMs from CMIP5 and 6. In this extensive metadata archive, you can see how
    the distinct modelling groups have built their models in terms of included sub-models / Earth System
    components.

What a great resource! It will certainly be widely used. We've added
reference to it here:

*L49-52: "Modeling centers often contribute several versions of their base
model under different names as well (Leduc et al. 2016); these variants
differ by, for example, the spatial resolution of some model components or
entire sub-models (see Brands et al. 2023), which may influence their
simulated climate in ways that are difficult to anticipate."*

3. Lines 56-57: „Adding further complexity, even uniquely named models from different modelling
   centers fall along a spectrum of uniqueness."

   I do not fully understand what you mean with this sentence. Could you provide an example for
   „uniquely named models from different modeling centers" ?

   That is a confusing formulation. We've revised it as:

   *L52-53: "Adding further complexity, models actually fall over a spectrum
   that ranges from effective replicates to fully independent entities."*

4. Lines 58-61: I think the Brands et al. (2023) GCM metadata archive is relevant in this sentence as well
   and the authors might wish to refer to it. The archive could be alternatively cited in lines 78-80 and is
   useful for determining „a priori" dependencies within in the CMIP ensemble, as defined by Boé
   (2018).

   We've added the reference as:

   *L53-55: "Different models share historical predecessors (Masson and Knutti,
   2011, Knutti et al. 2013), conceptual frameworks, and, in some cases,
   source code (Boé, 2018, Brands 2022b, Brands et al. 2023)"*

5. Lines 119-121. Meanwhile, the EURO-CORDEX model selection team has come to a final
   recommendation for the driving GCMs from CMIP6. Please see Sobolowski et al. (2023) for more
   details.

   Thank you for pointing me (Anna Merrifield) to this white paper! I am very
   happy to see that model independence is a key part of the EURO-CORDEX model
   subselection. A bit of a back story, this paper was largely prepared for
   submission in September 2022, but had to be "shelfed" for a few months for
   my maternity leave when my daughter arrived a few weeks early. While we
   hoped to have the paper out for consideration in the EURO-CORDEX selection
   process, time wasn't on our side this time. I hope the community will still
   find the method valuable!

   As we've worked to shorten the paper, we've removed reference to EURO-
   CORDEX.

6. Lines 205-217: Please indicate the time aggregation of the GCM and reanalysis data you are using. Is
   the study based on monthly-mean data?

   Absolutely, we've added reference to the monthly mean output here.

   *L183-188: For inclusion in Part I, the models also must provide (1) an
   estimate of ECS, calculated from a 4XCO$_2$ run using the Gregory method
   (Gregory et al., 2004) and (2) the following monthly-mean output fields
   (with their abbreviation and model output variable name given in brackets):
   near-surface 2-meter air temperature [SAT; tas], precipitation [PR; pr],*

*and sea level pressure [SLP; psl]. Further inclusion into Part II's*
*European case studies require the additional monthly-mean output fields of*
*sea surface temperature [SST; tos], and all sky and clear sky downwelling*
*shortwave radiation at the surface [rsds and rsdscs, respectively].*

7. Lines 218-219 and elsewhere: Would make sense to use the terms „a priori" and „a posteriori" model dependence (Boé 2017) in this study?

   We considered this and have used the „a priori" and „a posteriori"
   terminology in other papers (see Merrifield et al. 2020). Here due to there
   being several comparatives surrounding dependence for the reader to
   remember already (within-model vs. between-model, individual vs. family,
   etc.), we decided to formulate more as a model output-based independence
   definition with a metadata-based justification.

8. Lines 230-238: The definition of the INV and SME groups is clear but more information is needed on how you define the FAM group. For example, ACCESS-ESM1-5 is here considered an „SME" model, meaning that it „[...] is represented by multiple members (e.g., initial condition ensembles, perturbed physics ensembles, combinations thereof) but is not determined to be part of a broader multi-model family." However, a closer look at the „source" attributes of the corresponding netCDF files from ESGF and at the reference articles (doi: 10.1071/ES19035, 10.5194/gmd-12-4999-2019, 0.5194/gmd-4-723-2011,0.5194/gmd-4-1051-2011) reveals that the entire ACCESS GCM family is based on versions of the atmospheric sub-models (or AGCMs) developed at the MetOffice-Hadley Centre. Namely, ACCESS-ESM1-5 makes use of the „HadGAM2" AGCM that is also used by the HadGEM2-ES and HadGEM2-CC coupled model configurations. HadGAM2 was further developed into „MetUMHadGEM3-GA7.1", constituting the AGCM used in both Hadley Centre's and CSIRO's coupled model configurations used in CMIP6, e.g. HadGEM3-GC31-MM (doi: 10.1071/ES19040) and ACCESS-CM2 (doi: 10.1071/ES19040). Thus, it is reasonable to put the HadGEM and ACCESS coupled model configurations to the same family, as was done in Brands (2022), because they essentially share their atmospheric component. Following your nomenclature, this would mean assigning a „FAM" to ACCESS-ESM1-5. Note that all the aforementioned model metadata is available at one glance from Brands et al. (2023).

   This is an important point. Before we discuss though, we do contend that it
   is not ideal to have the FAM designation introduced before we describe how
   we make the FAM distinction. Because it serves as a good "quick reference",
   we plan to keep it in the table but highlight the preview aspect more in
   the text as:

   *L205-213: "Finally, to familiarize the reader with the concept of model*
   *families we will subsequently define, we also list the family group status*
   *of each model. The designation, "INDV", indicates a model is considered to*
   *be an individual represented by a single member. "SME" signifies that a*
   *model is represented by multiple members (e.g., initial condition*
   *ensembles, perturbed physics ensembles, combinations thereof) but itself is*
   *considered an individual entity. This means it was not found to be part of*
   *a broader multi-model family or "FAM" by the criteria we subsequently*
   *define. In total, the 218 CMIP6 simulations from 37 uniquely named models*
   *considered in Part I fall into 19 Groups (7 multi-model families, 8 single*
   *model ensembles, and 4 individuals) and the 75 CMIP5 simulations from 29*
   *uniquely named models fall into 20 Groups (8 multi-model families, 5 single*
   *model ensembles, and 7 individuals). In Part II, 197 CMIP6 simulations from*
   *34 uniquely named models and 68 CMIP5 simulations from 26 uniquely named*
   *models remain for the subselection exercise (Sup. Tabs. S1-S2)."*

   We initially assumed that ACCESS-ESM1-5 would be a part of the Met Office-
   Hadley Centre model family:

*L382-385: "We also anticipated three "extended" families based on an analysis of model metadata, summarized in Sup. Tabs. S1 and S2, and the work of Brands (2022b), which grouped models in CMIP5 and CMIP6 via shared atmospheric circulation error patterns. The first, shown in dark red (models 1-6) in Fig.3, is comprised of models with UK Met Office Hadley Centre atmospheric components."*

However, we found it did not meet our definition for family member:

*L385-392: "In CMIP6, intermember distances show five of the six models highlighted in red on the y-axis of Fig.3a, satisfy both the self-contained group and median intermember distance threshold criteria to form a family. This grouping makes sense as all five models (HadGEM3-GC31-MM, KACE-1-0-G, ACCESS-CM2, HadGEM3-GC31-LL, and UKESM1-0-LL) use the same MetUM-HadGEM3-GA7.1 atmospheric component (Sup. Table S1). The sixth model, ACCESS-ESM1-5, does not satisfy the self-contained criteria and is closer to other models in CMIP6 than it is to its anticipated family members. This likely occurs because ACCESS-ESM1-5 uses a CMIP5-era HadGAM2 atmospheric component rather than the CMIP6-era MetUM-HadGEM3-GA7.1 atmospheric component and highlights the potential for models in the same development stream to differentiate themselves from their successors."*

So the crux is: Can model development make a model functionally independent from a predecessor? This gets at what "independent" in this context means, which for us is a historically distinct enough simulation such that agreement in projection of future climate has meaning beyond "this model is agreeing with itself". An argument can be made that models developed by the bigger modelling centers like the Hadley Centre or NCAR conceivably could be functionally independent generation to generation. For example, from CAM4 to CAM5, NCAR updated many aspects of their aerosol and cloud parameterizations, alleviating several longstanding radiation biases (Kay et al. 2012).

Reference: Kay, J. E., and Coauthors, 2012: Exposing Global Cloud Biases in the Community Atmosphere Model (CAM) Using Satellite Observations and Their Corresponding Instrument Simulators. J. Climate, 25, 5190–5207, https://doi.org/10.1175/JCLI-D-11-00469.1.

(The qualifier "functionally" acknowledges that most models in CMIP are very similar to each other to start with regardless of origin, so an argument can be made that no current model is truly independent…)

To prime readers for the idea that model development could lead to independence, we've added to the opening paragraph of Section 3 Revisiting Model Dependence:

*L218–221: "In prior studies, it has been shown that a climate model's origins and evolution can be traced via statistical properties of its outputs (e.g. Masson and Knutti, 2011; Bishop and Abramowitz, 2013; Knutti et al., 2013). Output-based modelidentification can uncover hidden dependencies within the ensemble, e.g. models that are similar because they share components or lineages, but not names. The approach also has the advantage that it does not presume model similarity based on name alone; output from models in active development can evolve substantially from version to version (e.g. Kay et al., 2012; Boucher et al.,2020; Danabasoglu et al.,2020) while output from the same version of a model run at different modeling centers is often quite similar (Maher et al., 2021b)."*

9. Lines 240-247: Could you also shortly refer to the disadvantages of the a posteriori / output data–driven approach to measure GCM dependence ? Here, only the advantages are described so far.

This discussion was definitely missing. We've added reference to the primary disadvantage of the a posteriori / output data–driven approach to the opening paragraph of Section 3 Revisiting Model Dependence:

L221–225: "Risks arise, though, if model output used to determine similarity converges within a multi-model ensemble broadly, and thus becomes ineffective at differentiating between dependent and independent models (Brands, 2022b). To reduce the risk of similar output conflating dependent and independent models, we update the model dependence strategy from the ClimWIP independence weighting scheme (Brunner et al., 2020b) to revisit the concept of model families within CMIP."

And to the discussion:

L854–856: The potential for between-model convergence is cited as one of the primary drawbacks of using model output to determine dependence (Annan and Hargreaves, 2017; Brands, 2022b)."

10. Lines 259-261: Please add an equation to define inter-member GCM distance.

Absolutely. We've formally defined $I_{ij}$ here:

L237–245: "Intermember distance ($I_{ij}$) is calculated through pairwise RMSE between ensemble members i and j for each predictor field $\hat{y}$ individually. Individual predictor RMSEs ($\phi_{ij}$) are defined as:

$$\phi_{ij} = \sqrt{\frac{\sum_{k=1}^{p} w_k |\hat{y_i} - \hat{y_j}|^2}{\sum_{k=1}^{p} w_k}}$$

which reflects an RMSE weighted over the p gridpoints in a latitude / longitude domain, with $w_k$ indicating the corresponding cosine latitude weights. Each $\phi_{ij}$ is normalized by its respective ensemble mean value ($\bar{\phi}$) and all $\phi_{ij}$ are averaged together to obtain a single $I_{ij}$ for each member pair. As in Merrifield et al., 2020, $I_{ij}$ is comprised of two individual predictor fields, global-scale annual average SAT and SLP climatologies.

$$I_{ij} = \frac{1}{2}\sum_{l=1}^{2}\left(\frac{\phi_{ij}}{\bar{\phi}}\right)_l.$$ "

11. Lines 279-280: The observational density underlying theses gridded dataset is also reduced during the first half of the 20th century, particularly in the Southern Hemisphere.

This is a good point. We've amended the statement as:

L265–268: "However, we find that increasing the period back into the 19th century does not appreciably change intermember distances (not shown). Additionally, the 1905 start date may allow for backward-compatibility of the metric with future generations of CMIP should organizers decide to begin the historical period in the 20th century rather than the 19th century."

12. Lines 285-287: „relative change with respect to a historical period" is not considered „model performance", as far as I know. Traditionally, the term „model performance" refers to model error with respect to observations.

Thanks for bringing it to our attention, this sentence is confusing. What we were trying to convey is that using the absolute value of fields like SAT has traditionally been less common for model evaluation than using relative anomalies. We've revised the sentence to read:

*L274-275: "The absolute magnitude of a climatic field tends to be seen as secondary to its relative change with respect to a historical base period for most applications (Jones and Harpham, 2013)."*

13. Line 307: „confined to subtropical regions" > „confined to the tropics and subtropics"

Thank you, fixed as:

*L294–295: "This "low" between-model spread is largely confined to oceanic regions in the tropics and subtropics for both the SAT and SLP 1905-2005 climatologies."*

14. Lines 323-324: The Brands et al. (2023) metadata archive comprising names and versions of the sub-models used in each GCM configuration helps to identify the „very similar but differently named models" you refer to in this sentence.

We have also removed this sentence in our effort to shorten the paper, but reference to Brands et al. 2023 is made in several other places including:

*L425–426: "…or due to similar ocean component models (Brands et al. 2023)."*

*L51-52: "…for example, the spatial resolution of some model components or entire sub-models (see Brands et al. 2023), which may influence their simulated climate in ways that are difficult to anticipate."*

15. Figure 3b) I can here see 3 independent clusters instead of the 2 indicated in the caption.

Interesting, is it CanESM5 that you see as separate from the CMIP6 core visually? CanESM5 is on the independent side of CMIP6 core, about 6 units from its approximate center in the MDS projection while the MIROC models are about twice as far (~13 units). The broken axis may exacerbate this visual issue, but we feel it was necessary to allow readers to see how the bulk of the projection of the CMIP6 ensemble compared to the CMIP5 ensemble. To clarify, we've amended the figure caption to read:

*Figure 3: "Note that in panel b, a broken axis is used to emphasize the structure of the primary CMIP6 model core with respect to the independent constituents, MIROC6 and MIROC-ESL."*

16. Between page 16 and 17 it seems that some running text is missing.

Thank you for pointing this out. This may be a function of Figure 3 taking up all of page 16 so the running text jumps from page 15 to page 17. In the revision, we hope that this resolves based on our changes.

17. Lines 434-436: Similarities might be caused by the use of similar ocean models. Distinct versions of the same OGCM (NEMO) are used in the CNRM and IPSL GCMs (see Brands et al. 2023 for further details).

Thank you! We've added the reference as:

> L423–426: "Similarity in these cases cannot be traced to a particular
> atmospheric component model, but for CNRM and IPSL, similarity could have
> arisen through an effort to foster collaboration between the two French
> modeling groups after CMIP5 (Mignot and Bony, 2013) or due to similarities
> in ocean component model (Brands et al. 2023)."

**References**

Boé, J. (2018). Interdependency in multimodel climate projections: Component replication and result similarity, Geophysical Research Letters, 45, 2771– 2779. https://doi.org/10.1002/2017GL076829

Brands, S. (2022). Common error patterns in the regional atmospheric circulation simulated by the CMIP multi-model ensemble. Geophysical Research Letters, 49, e2022GL101446. https://doi.org/10.1029/2022GL101446

Brands, S., Tatebe H., Danek, C., Fernández, J., Swart, N. C., Volodin, E., Kim, Y.H., Collier, M., Bi, D., Tongwen, W. (2023). SwenBrands/gcm-metadata-for-cmip: First standalone version of GCM metadata archive "get_historical_metadata.py" (v1.1). Zenodo. https://doi.org/10.5281/zenodo.7813495

Sobolowski, S. et al. (2023). EURO-CORDEX CMIP6 GCM Selection & Ensemble Design: Best Practices and Recommendations. Zenodo. https://doi.org/10.5281/zenodo.7673400

---

## Author Comment (AC3)

This manuscript introduces a method (ClimSIPS) which select subsets of CMIP models based on model independence, model performance and spread. In the second part the authors describe a case study for European summer and winter.

The manuscript fits the scope of GMD and is very helpful by dealing with the large ensemble of CMIP5 and especially CMIP6 models. Additionally, the change of the ECS distribution when considering only one model family member is very interesting. Nevertheless, the manuscript has reached an extreme length and is written very detailed. I am wondering if there are places where the text could be shortened.

Thank you for your review! We really appreciate your takeaways from what we agree is a bit of an interminable read. Following your feedback, we've made some substantial cuts to the article, listed as follows:

- Moved CMIP5 subselection (Figure 11) to the supplement and removed discussion of EURO-CORDEX as a benchmark from the main text
- Shortened the paragraphs on robustness and model agreement in Section 1.1
- Removed lists of initial condition ensembles used in the construction of the intermember distance metric from the main text
- Shortened discussion of within-model vs. between-model spread masking
- Reworded sentences to be more concise throughout
- Improved mathematical notation and added equations to be more precise with the cost function terms.

In total, we have reduced the length of the paper by several pages, even with additions requested during the review period.

Some small comments:

Line 53: Capital letter in the beginning of the sentence: "Modeling centers…"

Thank you; fixed.

Line 86/87: A verb is missing in the first part of the sentence.

Thank you for the catch. We've revised as:

L72–73: "Initial versions of ClimWIP based performance and independence definitions on the same set of predictors, which lead to concerns about convergence to reality."

Figure 1: The quality of the figure is quite bad and difficult to read.

That's very good to know, thank you! We've increased the dpi of the png image from 300 to 800 to rectify this.

Line 568: Is there a special reason for choosing this reanalyse dataset?

There was not a special reason for using the observational datasets we used to demonstrate the method. The reanalysis datasets were chosen based on availability at the time of method development under the assumption that we would perform further sensitivity testing to account for observational uncertainty. As we continued to develop the method, though, it became clear that this exact definition of performance was becoming a tangential avenue of inquiry. In

ClimSIPS, the performance metric is the simplest to swap out; models simply need a scalar rank, which can be obtained in many different and interesting ways. We envision most users will want to define their own performance metrics and therefore decided to focus method evaluation energy elsewhere (e.g., on automating selection of the spread-maximizing ensemble member from each SME).

To address this in the paper, we've added the following:

*L560–562: "We found using a single observational estimate for each predictor to be sufficient for demonstrating ClimSIPS; the method's sensitivity to representations of observational uncertainty, different predictor combinations, and alternative performance definitions all warrant further exploration."*

Line 945 and 947: Is this grade of precision of the numbers really needed here?

This is a good point. It is definitely not needed for the combinatoric explosion argument. We've amended the sentences to read:

*L920–923: "34 choose 5 subselection iterated over more than 16 billion cost function values, which, run in parallel, took approximately two hours to run on 24 cores. Not evaluated here, 34 choose 10 subselection, with the cost function computed $6.6 \times 10^{12}$ times, would take considerably longer, an estimated three weeks to run, even in parallel on 48 cores."*

---

## Author Response (AR2)

Thank you for carefully editing our manuscript. Following your suggestions, we have fixed the mis-formatted reference, added "v1.0.1" to the title, and removed the paragraph from the abstract.  Please let us know if anything else comes up!